# FROM COMMANDS TO PROMPTS: LLM-BASED SEMANTIC FILE SYSTEM FOR AIOS

**Zeru Shi**♠ *, **Kai Mei**♠*, **Mingyu Jin**♠, **Yongye Su**♡, **Chaoji Zuo**♠, **Wenyue Hua**♠,
**Wujiang Xu**♠, **Yujie Ren**‡, **Zirui Liu**§, **Mengnan Du**◇, **Dong Deng**♠, **Yongfeng Zhang**♠†
♠ Rutgers University ♡ Purdue University ◇ New Jersey Institute of Technology
‡ EPFL § University of Minnesota

## ABSTRACT

Large language models (LLMs) have demonstrated significant potential in the development of intelligent LLM-based agents. However, when users use these agent applications to perform file operations, their interaction with the file system still remains the traditional paradigm: reliant on manual navigation through precise commands. This paradigm poses a bottleneck to the usability of these systems as users are required to navigate complex folder hierarchies and remember cryptic file names. To address this limitation, we propose an LLM-based Semantic File System (LSFS) for prompt-driven file management in LLM Agent Operating System (AIOS). Unlike conventional approaches, LSFS incorporates LLMs to enable users or agents to interact with files through natural language prompts, facilitating semantic file management. At the macro-level, we develop a comprehensive API set to achieve semantic file management functionalities, such as semantic file retrieval, file update summarization, and semantic file rollback). At the micro-level, we store files by constructing semantic indexes for them, design and implement syscalls of different semantic operations, e.g., CRUD (create, read, update, delete), group by, join. Our experiments show that LSFS can achieve at least 15% retrieval accuracy improvement with $2.1\times$ higher retrieval speed in the semantic file retrieval task compared with the traditional file system. In the traditional keyword-based file retrieval task (i.e., retrieving by string-matching), LSFS also performs stably well, i.e., over 89% F1-score with improved usability, especially when the keyword conditions become more complex. Additionally, LSFS supports more advanced file management operations, i.e., semantic file rollback and file sharing and achieves 100% success rates in these tasks, further suggesting the capability of LSFS. The code is available at https://github.com/agiresearch/AIOS-LSFS.

## 1 INTRODUCTION

In recent years, researchers have put great efforts in integrating AI to provide better services for serving applications. For example, machine learning algorithms have been studied to optimize system resource allocation and and improve system efficiency (Blair et al., 1987; Schneider et al., 2020; Gong et al., 2024). The emergence of large language models (LLMs) has further catalyzed the integration of AI into serving applications. The great reasoning and planning ability of LLMs facilitates the development of LLM-based agents, including single-agent applications (Yang et al., 2024b; Zhang & Zhang, 2023; Gur et al., 2023; Deng et al., 2024) and collaborative multi-agent applications (Wu et al., 2024a; Ge et al., 2024; Shen et al., 2024; Hong et al., 2023).

In these LLM-based agents, file management operations primarily rely on the traditional way and traditional file systems primarily rely on file attributes to build metadata. These attributes, typically obtained by scanning the file, include file size, creation and modification timestamps. The actual file content is stored as binary data, with traditional file systems leveraging index structures such as B+ trees to efficiently locate this data. While these designs continue to evolve and improve, they

---

* Equal contribution and shared co-first authorship
† Corresponding author.

generally overlook the semantic content information within files, making it difficult for traditional file systems to support tasks that require deeper semantic understanding. It is unable to leverage the high-level semantic meaning in the natural language context. ❶ For instance, if two files have similar content which cannot be distinguished by simple string matching, traditional file systems lack the ability to organize or retrieve these files based on content similarity. ❷ User interactions with traditional file systems require complex operating system commands or manual navigation through the user interface, forcing users to precisely recall file names or locations. For systems with numerous files, this retrieval process can be inefficient and time-consuming, reducing overall system usability. Nowadays, based on the strong language understanding capability of LLMs, we can make better use of the file content and semantic information for file management by introducing LLMs into the system. However, existing works on using LLMs to facilitate file management are mostly conducted on the application level, which targets at designing specific agent for file retrieval and manipulation (Liu et al., 2024; Talebirad & Nadiri, 2023). The community still lacks a more general LSFS to serve as a common foundation that can be used by various agents on the application-level. Mei et al. (2024) have proposed the AIOS, a foundational architecture for serving LLM-based agents. On top of AIOS, we propose LLM-based Semantic File System (LSFS) to more effectively integrate LLM and traditional file system to provide fundamental semantic file management services for AIOS.

For problem ❶, our LSFS introduces a semantic-based index structure that leverages a vector database for file storage. By extracting semantic features from the file content and generating corresponding embedding vectors, LSFS incorporates semantic information into its file operations. Additionally, we have designed numerous reusable syscall interfaces for LSFS, modeled after traditional file system functions. At the same time, we design several APIs that can realize complex file functions based on the syscalls. These syscalls and APIs not only can realize the basic functions of the file system but also can provide the operations that the traditional file systems do not include.

To address problem ❷, we integrate LLM into the API for complex functions and introduce a template system prompt. This allows us to utilize LLM to extract keywords from the natural language of user input and map them effectively as API calls or syscalls, streamlining the interaction between users and the system. The comparison of commands executed by users in traditional file systems and LSFS is shown in Figure 1. In Figure 1(a), When a user wishes to modify the content of files, they must input a

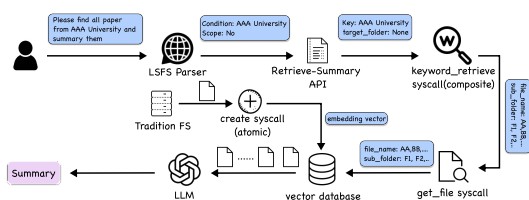

Figure 1: A fine-grained example of the pipeline of changing file of traditional system and our LSFS.

specific command in the terminal, requiring them to remember the correct operator and the exact paths for both the target and source files, placing heavy burdens on users. However, LSFS can effectively solve this problem. Users only need to manage files by typing a natural language prompt as a command. For example, as shown in Figure 1(b), the user just needs to input a simple natural language description, our LSFS is able to understand prompts and perform the corresponding operation, which greatly simplifies the operation complexity. Furthermore, to reduce the hallucination problem of generating inaccurate instructions of LLM, especially those irreversible operations, we design systematic safety insurance mechanisms in LSFS, such as safety checks for irreversible operations and user verification before instruction execution.

Overall, our research contributes as follows:

- We introduce an **LLM-based Semantic File System (LSFS)** to manage files in a semantic way. By altering the file storage structure and method, LSFS incorporates the semantic context of files, optimizing the fundamental functions of traditional file systems. Additionally, we develop a variety of reusable syscalls and APIs within LSFS, allowing for extended functionality and enabling future developments based on this system.

- Our system designs a **LSFS parser**, which can parse natural language prompts into executable APIs supported by LSFS, enabling the execution of relevant file management tasks. This allows users to control and manage files using simple natural language prompts, acting as a bridge that translates user/agent instructions into system actions.

- To avoid unintended operations in LSFS, especially those irreversible operations, we design systematic **safety insurance mechanisms**, such as safety checks for irreversible operations and user verification before instruction execution, ensuring the safety and accuracy of LSFS.

- Through our experiments, we validate the completeness of functions of LSFS, while also evaluating the performance of LSFS in various file management tasks. Our experiments show that LSFS performs better in semantic file management tasks, e.g., achieve at least 15% retrieval accuracy improvement with $2.1\times$ higher retrieval speed in the semantic file retrieval. Besides, LSFS also maintains good functionality in traditional file management tasks, i.e., keyword-based file retrieval and file sharing, with usability improvement.

## 2 RELATED WORK

### 2.1 SEMANTIC FILE SYSTEM

Currently, file storage and retrieval primarily rely on an index structure maintained by the system, where file metadata points to the location of file on the disk (Dai et al., 2022). While optimizing the index structure can enhance retrieval efficiency, the current storage model is still largely dependent on the keywords extracted from the content of file. Gifford et al. (Gifford et al., 1991) were the first to propose a semantic file system, which introduced a layer that generates directories by extracting attributes from files, enabling users to query file attributes through navigation. (Eck & Schaefer, 2011) proposed a semantic file system to manage the data. Many subsequent works have integrated semantics into metadata (Hua et al., 2011; Mahalingam et al., 2003; Hua et al., 2009; Hardy & Schwartz, 1993; Mohan et al., 2006). (Hua et al., 2013) utilized semantics to reduce the relevance of queries using the semantic similarity between files based on the semantic naming system. Leung et al. (Leung et al., 2009) used semantic information combined with the file system design of graphs to provide scalable search and navigation. On the other hand, Bloehdorn et al. (Bloehdorn et al., 2006) proposed to manage files through semantic tags. Schandl et al. (Schandl & Haslhofer, 2009) developed an approach for managing desktop data using a semantic vocabulary. In contrast, our semantic file system is based on the strong language understanding ability of LLMs. Besides, it integrates comprehensive semantic information across all aspects of the file system–from storage and file operations to practical applications. This holistic approach enhances the ability of system to understand and manage files, significantly improving functionality beyond what traditional systems and earlier semantic file systems offer.

### 2.2 SEMANTIC PARSER

Researchers have also devoted efforts to developing semantic parsers (Kamath & Das, 2018; Mooney, 2007; Wong & Mooney, 2006; Clarke et al., 2010; Yih et al., 2014; Jin et al., 2025a) capable of transforming natural language into a machine-interpretable format. Iyer et al. (Lawrence & Riezler, 2018) subsequently focused on parsing database commands, while Berant et al. (Berant et al., 2013) proposed a question-answer pair learning approach to enhance parsing efficiency. In further work, the same authors explored a paraphrasing technique (Berant & Liang, 2014) to improve semantic parsing performance. Poon et al. (Poon & Domingos, 2009) introduced a Markov logic-based approach, and Wang et al. (Wang et al., 2015) addressed the challenge of building parsers from scratch in new domains. Ge et al. (Ge & Mooney, 2005; Jin et al., 2025b) proposed a parse tree-based method for more accurate semantic analysis. Some paper using the long CoT to make LLM parser the sentence(Jin et al., 2024b;a). Notably, Lin et al. (Lin et al., 2018) were the first to integrate a semantic parser into an operating system, leveraging a dataset of bash commands and expert-written natural language to establish a mapping between the two. However, this approach faced limitations in handling complex semantics and unseen natural language. In contrast, our LSFS is built upon constructing semantic indexes for files in the format of embedding vectors, improving its generation ability to understand and process diverse natural language inputs.

### 2.3 OS-RELATED LLM-BASED AGENTS

The power of LLMs have fostered the development of LLM-based agents in many fields, including chatbox (Achiam et al., 2023; Team et al., 2023; Guo et al., 2025; Yang et al., 2024a), code assistant (Wei et al., 2023; Hui et al., 2024; Nijkamp et al., 2023; Zhu et al., 2024) and recommender systems (Xu et al., 2025; Zhang et al., 2024; Wang et al., 2023). Recent works (Yang et al., 2024b; Qian et al., 2023; Wu et al., 2024b; Bonatti et al., 2024; Wang et al.; 2024; Yang et al., 2023) focus on leveraging LLM-based agents to solve OS-related tasks. To help users solve more practical OS-related tasks

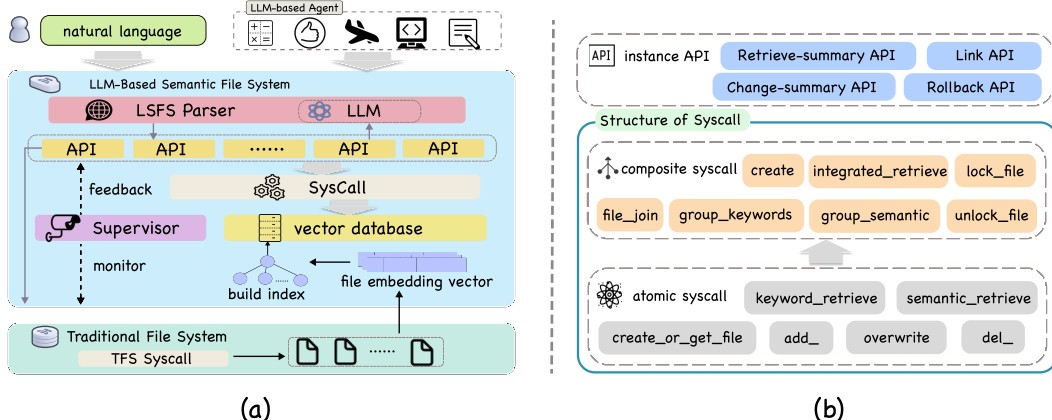

Figure 2: (a) provides a overview of the LSFS architecture, and (b) shows the internal APIs and syscalls in LSFS.

with natural language interaction, different agents are proposed for both PCs (Wu et al., 2024b; Bonatti et al., 2024) and mobile devices (Wang et al.; 2024; Yang et al., 2023). Wu et al. (Wu et al., 2024b) developed LLM-based agents for co-piloting users' interaction with computers, such as drawing charts and creating web pages. MetaGPT (Hong et al., 2023) employs a sophisticated large language model in a multi-agent conversational setting to automate software development, assigning specific roles to various GPTs for seamless collaboration. Beyond the application-level research on LLM-based agents, researchers also explored integrating LLMs into the system-level (Mei et al., 2024; Bonatti et al., 2024), which target at low-level and general management services (e.g., scheduling and resource allocation) for agent applications running on the top. While researches of agent systems primarily focus on build of LLM applications that can leverage file resources, our represents a fundamental innovation in the infrastructure that manages file resources based on semantics to support LLM-based agent systems.

# 3 ARCHITECTURE OF LSFS

**Design considerations.** Before delving into the architecture of our LSFS, we outline several key considerations that guided its design. ***Isolation and modularization:*** A layered architecture can better separate concerns and assign distinct responsibilities to each layer. This isolation can reduce the complexity of individual components and enable each layer to evolve independently without introducing unintended dependencies. A modular approach to build components in this system also enables individual components to be easily modified or scaled. In this way, it supports flexibility, enabling replacement, optimization, or extension of individual modules without requiring significant changes to the overall system. ***Performance and Efficiency:*** To ensure the performance of the system, streamlined data flow mechanisms are necessary. Each stage of the pipeline is carefully designed to handle data transformations efficiently. Besides, to improve efficiency, each component of the system should consider the lightweight choice and operations between different components should be decoupled to ensure parallelism. processing. ***Fault Tolerance and Reliability:*** Fault-tolerant mechanisms are necessary to ensure uninterrupted operation, for example, the rollback mechanism to reverse mistaken operations or recover from unexpected errors, improving overall reliability.

**Overview of the architecture.** Under the above considerations, we present the design of our LSFS. Figure 2(a) outlines the overall architecture of our LSFS. LSFS operates as an additional layer on top of traditional file systems, working as a bridge between agents/users and traditional file systems. We leverage the layered architecture with segregated LSFS APIs and LSFS syscalls so that APIs can focus on aligning with natural prompts while LSFS syscalls focus on aligning with low-level operations over files and databases. To build semantic index for files in the LSFS, we leverage an a lightweight embedding model, i.e., all-MiniLM-L6-v2 (Reimers & Gurevych, 2019), commonly used in vector databases, thereby supporting more advanced file operations which requires semantic understanding of file content. LSFS includes a supervisor that monitors changes in the traditional file system and synchronizes them with LSFS in real time. This synchronization, combined with the rollback mechanism, ensures fault tolerance and maintains consistency between LSFS and the

underlying file system. Figure 2(b) presents an overview of the syscall structure in LSFS, which contains three parts and we will elaborate in Section 4.1.

# 4 IMPLEMENTATION OF LSFS

In this section, we introduce our implementation of the LSFS. We present the key functions implemented in our LSFS and compare the counterparts with traditional file systems, which can be seen from the Table 1. We introduce the implementation of LSFS from the bottom to the top in

Table 1: Comparison of some key functions between our LSFS and traditional file system (TFS).

| Function | Implementation in TFS | Implementation in **LSFS** |
|---|---|---|
| create new directory | `mkdir()` | `create()` |
| create file | `touch()` | `create_or_get_file()` |
| open file | `open()` | `create_or_get_file()` |
| read file | `read()` | `create_or_get_file()` |
| get file state and metadata | `stat()` | `create_or_get_file()` |
| delete directory | `rmdir()` | `del_()` |
| delete file | `unlink()` / `remove()` | `del_()` |
| write data | `write()` | `add_()` |
| overwrite data | `write()` | `overwrite()` |
| update the access time | `utime()` | `update_access_time()` |
| automatic comparison | — | `compare_change()` |
| generate link | `symlink()` / `link()` / `readlink()` | `generate_link()` |
| lock or unlock file | `flock()` | `lock_file()` / `unlock_file()` |
| rollback | `snapshot` + `rollback` | `rollback()` |
| file group | — | `group_keywords()` / `group_semantic()` |
| merge file | `cat` | `file_join()` |
| keyword retrieve | `grep` | `keyword_retrieve()` |
| semantic retrieve | — | `semantic_retrieve()` |
| hybrid retrieval | — | `integrated_retrieve()` |

the LSFS architecture shown in the Figure 2. In the following parts, we start by introducing the basic syscalls implemented in LSFS and introduce the supervisor which interacts between LSFS syscalls and traditional file systems. Then we present the APIs that built upon the syscalls to achieve more complex functionalities. After that, we introduce the LSFS parser on top to show how natural language prompts have been decoded into executable LSFS APIs. At last, we use different concrete prompts to show how different modules in the LSFS are executed to achieve functionalities.

## 4.1 BASIC SYSCALL OF LSFS

In this section, we introduce the syscalls implemented for LSFS. These syscalls are primarily categorized into two types: atomic syscalls and composite syscalls. Atomic syscalls involve operations covering the most basic operations, e.g., create, retrieve and write of files. Composite syscalls are combinations of two or more atomic syscalls to execute composite functions, e.g., join and group by. A comparison of the operational complexity of LSFS and traditional operating systems is shown in the Table 5. This section shows the differences in detail with a few commands.

**Atomic Syscall of LSFS.** These syscalls involve the atomic operations that cannot be divided further into sub operations, i.e., creation, retrieval, write, and deletion of files.

- **create_or_get_file()** This syscall integrates various functions of traditional file systems, including creating, reading, and opening files, and performs specific operations based on the provided parameters. The return value of this syscall can be used to retrieve file metadata, modification timestamps, the file's memory path, and other essential information.
- **add_()** This syscall is used to write new content to the end of a specified file within the LSFS.
- **overwrite()** This syscall is used to overwrite the contents of the original file with the new file and generate new metadata for this file as required by the user.
- **del_()** This syscall is designed to delete specified files and offers two methods of deletion. First, it allows deletion by specifying the file name or file path. Second, it supports keyword-based deletion, identifying and removing files that contain a given keyword. Additionally, if all files within a directory are deleted, the syscall automatically remove the directory itself.
- **keywords_retrieve()** This syscall is used to implement a keyword search function that retrieves files containing a keyword in a specified directory. It supports single condition matching and multi-condition matching, and returns the filename and file contents.

- **semantic_retrieve()** This syscall is used to implement the semantic matching function to retrieve the top-n highest semantic similarity files in directory and retrieval conditions according to the similarity score. It returns the filename and file contents.

**Composite Syscall of LSFS.** These syscalls involve composite operations that are built by combining two or more atomic syscalls to perform operations.

- **create()** This syscall is used to create files in bulk in LSFS by importing the path of the folder in memory and importing all the files in the folder under the corresponding directory.
- **lock_file() / unlock_file()** The two syscalls are used to lock/unlock a file by changing the file state to read-only via *lock_file* and changing the file permission to read-write via *unlock_file*.
- **group_semantic()** The syscall can select the content in the specified directory and retrieve the files that have high similarity with the query, create a new directory, and place the selected files in the directory to facilitate the operation of the files that have the same subject.
- **group_keywords()** This syscall can select the files that contain the retrieved keywords in the specified directory, create a new directory, and place the selected files in the directory to facilitate the operation of the files that contain the same keywords.
- **integrated_retrieve()** This syscall combines two retrieval methods to retrieve the files that contain a particular keyword and that are similar in content to the retrieval query. The order of retrieval is keyword search first, and then semantic search.
- **file_join()** This syscall can be used to concatenate two files into a single file, either by creating a new file to concatenate or by concatenating the original file directly.

## 4.2 SUPERVISOR

The supervisor is implemented to track the changes in the files in the disk and sync the changes to the LSFS. The supervisor periodically scans the files within its specified directory. When it detects any change or deletion of the file content, it automatically synchronizes this information with the LSFS by invoking the appropriate syscall. This ensures that the state of the file in the LSFS reflects the current state of the file in memory. LSFS also leverages the process lock mechanism to ensure that multiple processes can access the file correctly without synchronization problem.

The supervisor also supports the change log, for example, when a file is modified, the supervisor invokes the LLM to generate a detailed modification log, compares the contents of the file before and after modification.

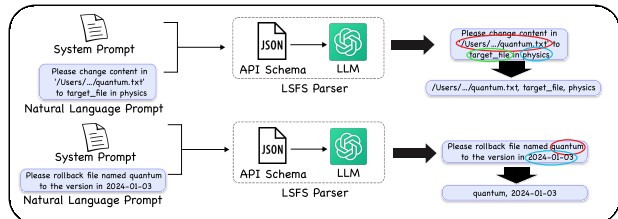

Figure 3: The example of using LLMs to extract the key information from natural language prompt.

## 4.3 API OF LSFS

In this module, we introduce the APIs that are implemented on top of the syscalls mentioned in Section 4.1 to support higher-level semantic file management functions. Specifically, we provide the following APIs that cover the basic semantic file management requirements, i.e., **semantic CRUD** (create, read, update, and delete) of files. The details of APIs are presented in Appendix B.

- **Retrieve-Summary API.** This API implements the retrieval operations in LSFS, including keyword search and semantic search, and feeds back the retrieved content to the user through LLM.
- **Change-Summary API.** This API implements the modification of the file content of the object file in LSFS, and it also can helpcompares the contents of the file before and after modification through LLM and gives a summary of the change.
- **Rollback API.** This API allows you to rollback a given file and provides several ways to do so, which includes rollback by date or rollback by version number.
- **Link API.** This API generates a shareable link for a given file. Users can set an expiration date for the link, after which the link will be invalid.

We also design and implement the LSFS parser to parse natural language prompts into API calls that can be executed in the LSFS, which will be introduced in Section 4.4.

## 4.4 LSFS Parser

To parse natural language prompts into executable API calls, we implement a LSFS parser based on the LLMs and designs well-structured json-format schemas for each API inside the parser. Previous works explore the parser to parse natural language into well-structured data (Kamath & Das, 2018; Mooney, 2007; Wong & Mooney, 2006). Recently, related studies tend to explore LLMs for this translation (Mior, 2024; Chen et al., 2024; Wu et al., 2024b) Inpired by these works, we design the LSFS parser that parse natural language into well-structured json data. Without specific mention, our parser is based on GPT-4o-mini by default, evaluations of using other LLMs will be reports in Section 5.1. By leveraging this parser, natural language prompts can be parsed into executable API calls (i.e., API function names and API function arguments), enabling seamless execution of the API command. This can help deal with natural language prompts with multiple and complex situations, benefiting interactions between natural language prompts and LSFS. As illustrated in the accompanying Figure 3, when input alongside the command of user, the LSFS parser is able to extract the key parameters, including function names and arguments in a comma-separated format.

## 4.5 The Interaction between Modules

In Figure 5, we present the examples to demonstrate how components of LSFS interact with each other to achieve different functionalities. The upper section of Figure 5 depicts the workflow of the *retrieve-summary API*, while the lower section outlines the workflows of the *change-summary API* and *rollback API*. In the upper part of Figure 5, the LSFS parser decodes prompts into API calls with API name and API arguments. Then LSFS executes the API to check vector database to get results. We used llamaindex to index the database and

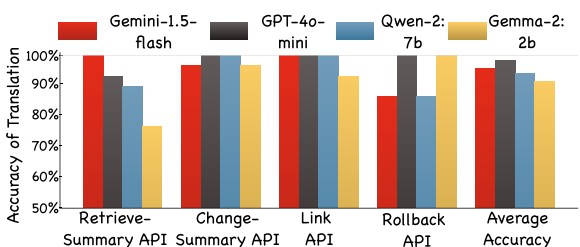

Figure 4: The accuracy of LSFS parser in translating natural language prompt to executable API calls.

subsequently retrieved the contents of the vector database by llamaindex. This API also provides user-interaction interface for the users to verify results. After the verification, the content will be summarized by leveraging LLM. In the lower part of Figure 5, when a modification request is submitted, the LSFS parser decodes the file information (name and location) that is to be modified. The LSFS then modifies the semantic changes in both the vector database and the files stored in the disk. Meanwhile, the supervisor of the LSFS is kept running to ensure consistency between the semantic index of files in the LSFS and the files stored in the disk. Upon updated, the summarization API compares the file contents before and after the change to generate a detailed change log. Additionally, the API stores the pre-modification content in the version recorder. If a rollback is requested, the API retrieves the specific version from the version recorder and synchronizes it in both the LSFS and files in the disk to keep the versions in sync between the two systems above.

## 5 Evaluation

In this section, we propose the following research questions regarding the performance of LSFS and conduct experiments to answer these research questions.

- **RQ1:** What is the success rate of the LSFS parser to parse natural language prompts into natural words which can map into the parameters and make API calls executable?

- **RQ2:** How does LSFS perform in semantic file management tasks?

- **RQ3:** Can LSFS still maintain good performance in non-semantic file management tasks?

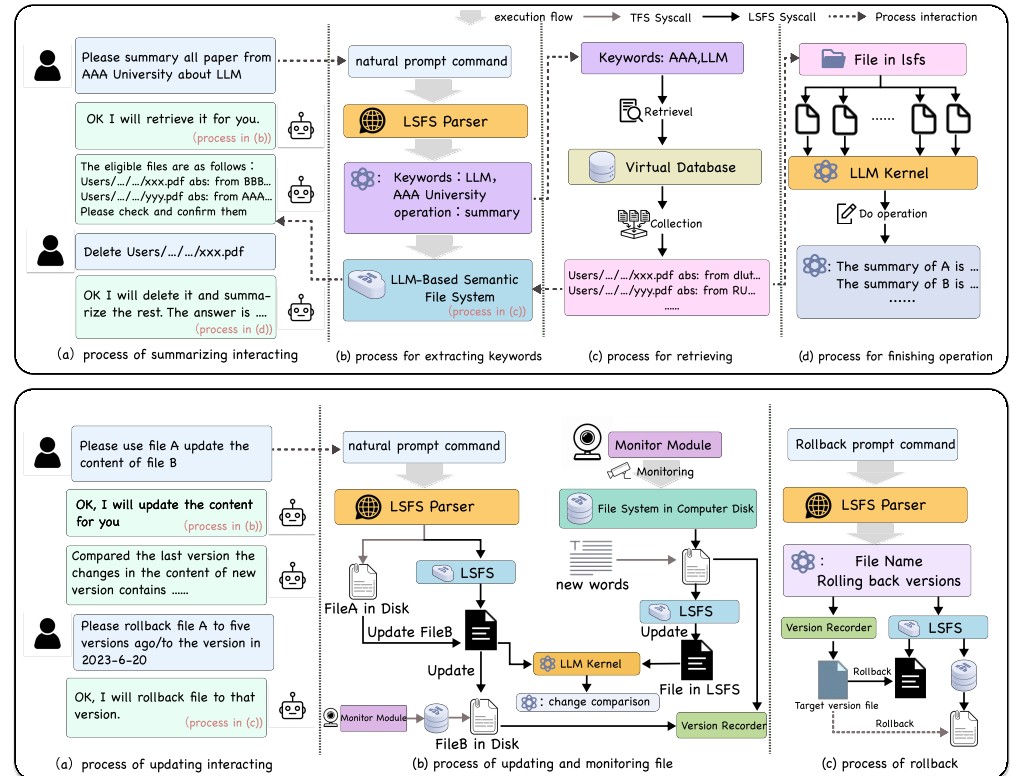

Figure 5: Details of different API callings inside the LSFS. In this figure, (a)-(d) show interactive examples of how LSFS solves file management tasks step by step.

## 5.1 RQ1: EFFECTIVENESS OF LSFS PARSER

For RQ1, we assess the accuracy of the LSFS parser in translating user natural language prompt into executable LSFS API calls. We evaluate the accuracy of LSFS parser with 30 different samples for each API on different LLM backbones, i.e., Gemmi-1.5-Flash, GPT-4o-mini, Qwen-2, and Gemma-2. The results, illustrated in Figure 4, reveal that the LSFS parser performs exceptionally well on parsing prompts related to *change-summary API* and *link API* (for which the semantic information in the user prompt is relatively simple), achieving higher accuracy across all LLMs (i.e., over 90%), where GPT-4o-mini and Qwen-2 all reach 100% accuracy. For more complex prompts, such as those intended for the *rollback API* and *retrieve-summary API* (for which the semantic information in the user prompt is complex), accuracy remains above 85% for most models, except for *Gemma-2*.

The average parsing accuracy reaches 90%. These results show that the LSFS parser can effectively parse natural language information into executable API calls, showcasing its reliability in diverse scenarios. For safety consideration, in all cases, the parsed API calls are provided to users for confirmation and approval before execution, avoiding irreversible file operations like deleting files or directories. More experimental results are in the Section H.1.

## 5.2 RQ2: ANALYSIS OF LSFS IN SEMANTIC FILE MANAGEMENT TASKS

To answer RQ2, we evaluate the performance LSFS on semantic file management tasks.

**Performance Analysis in *Semantic File Retrieval*.** In our experiments, we compare the performance of using LSFS and without using LSFS under the same LLM backbone. The details of the prompts we use for the comparison are in the Appendix D. Specifically, we use Gemini-1.5-flash and GPT-4o-mini as the LLM backbone, respectively, for the comparison. We don't use Gemma-2 and Qwen-2 because the unstable performance of them, this instability makes it challenging to assess the model's reliability and to derive meaningful conclusions from the system's performance. As

Table 2: Comparison of the accuracy and execution time between using LSFS and the baseline which incorporates LLM into traditional file system without using LSFS.

| LLMs backbone | # files | Accuracy of target file retrieval | | Retrieval time | |
|---|---|---|---|---|---|
| | | w/o LSFS | **w/ LSFS** | w/o LSFS | **w/ LSFS** |
| Gemini-1.5-flash | 10 | 75.0% | 95.0%(20.0%↑) | 97.40(s) | 14.39(s)(85.2%↓) |
| | 20 | 77.3% | 91.3%(14.0%↑) | 213.69(s) | 16.69(s)(92.2%↓) |
| | 40 | 70.91% | 93.4%(22.5%↑) | 312.39(s) | 23.86(s)(92.4%↓) |
| | 120 | 35.2% | 92.9%(164%↑) | 605.59(s) | 48.08(s)(92.1%↓) |
| GPT-4o-mini | 10 | 80% | 95.0%(15.0%↑) | 61.14(s) | 30.64(s)(49.9%↓) |
| | 20 | 69.1% | 91.3%(22.2%↑) | 129.92(s) | 40.39(s)(68.9%↓) |
| | 40 | 69.2% | 93.4%(24.2%↑) | 239.49(s) | 57.1(s)(76.2%↓) |
| | 120 | 63.8% | 92.9%(45.6%↑) | 938.68(s) | 88.93(s)(90.5%↓) |

shown in Table 2, using LSFS to implement the retrieval function significantly enhances both the accuracy and the efficiency compared to only leveraging LLM for retrieval without using LSFS. As file number increases, the retrieval accuracy tends to drop significantly when using LLM for retrieval without LSFS. This is because more files can lead to longer context for the LLM, which degrades the performance of LLM of identifying information in the long context. By contrast, using LSFS can still achieve good retrieval accuracy and have much better retrieval efficiency when file number increases, because lsfs replaces the reasoning process of LLM by using keyword matching and semantic similarity matching, it saves a lot of time and avoids the errors of LLM when facing complex input text. The example of API with LSFS and API without LSFS are in Appendix I.

**Scalability Analysis in *Semantic File Rollback*.**

LSFS supports semantic file rollback, which enables the restoration of a file to a particular version specified by the time requested by the user or number of versions, recorded by the Version Recorder. We vary the the number of rollback versions and calculate their corresponding rollback time, to evaluate the stability and efficiency of the version rollback process, The results, shown in Figure 6, illustrate the consistency in the time consumed during version rollbacks. We use Gemmi-1.5-Flash, GPT-4o-mini, Qwen2 as the LLM backbones for the experiments. In our experiment, we rollback file with versions the range from 5 to 40, using increments of 5. Each rollback is simultaneously updated in both the LSFS and file stored in the disk. As shown in the Figure 6, across all three LLM backbones, the rollback time

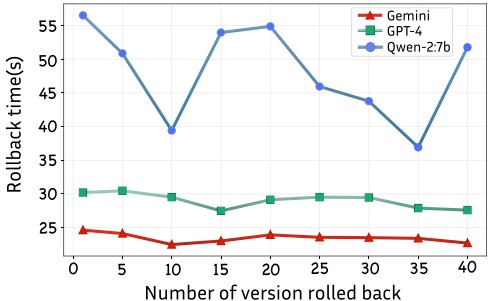

Figure 6: The relationship between the number of versions of a rolled back file and the rollback time.

does not increase exponentially with the number of versions rolled back. Instead, it tends to plateau with a stable rollback time < 1 min even if file number increases, suggesting the scalability of semantic file rollback supported by LSFS.

**Effectiveness of *Supervisor*.** In order to evaluate the effectiveness of our supervisor and prove that with the increase of the number of management files required, the response time required by supervisor does not increase exponentially when files are updated, we conducted the following experiment to obtain the response time by continuously increasing the number of management files. The experimental results are shown in the 3: Meanwhile, CPU usage is maintained between 0.1%

Table 3: Time comsuming supervisor with the increasing of file number

| # files | 100 | 200 | 400 | 800 | 1600 |
|---|---|---|---|---|---|
| Response time(ms) | 0.6(ms) | 1.1(ms) | 2.1(ms) | 4.2(ms) | 4.4(ms) |

and 0.2%. These results show that the supervisor is efficient, with millisecond-level response times and low CPU usage remained, even as the number of files increases.

### 5.3    RQ3: ANALYSIS OF LSFS IN NON-SEMANTIC FILE MANAGEMENT TASKS

For RQ3, we evaluate on non-semantic file management tasks to measure whether LSFS can still maintain good performance as traditional file systems in these tasks.

**Performance Analysis in *Keyword-based File Retrieval*.**    In this section, we compare LSFS and traditional file system in keyword-based file retrieval task. The task is to use keywords existing in the filename or file content to retrieve files. We build a hierarchical file folder with file numbers as 10, 20, and 40, respectively, for this task. We use two types of retrieval prompts, i.e., single-condition and multi-condition, to evaluate LSFS and traditional file systems to retrieve relevant files containing specific keywords. The details of how we construct prompts for this task can be seen at Appendix D. We consider the following methods as the retrieval baselines in the traditional file system. It is important to note that the original *grep* command can only deal with plain text files, such as *.txt* and *.md* files, and cannot support binary files, such as *.pdf* and *.doc* files. Therefore, we make two enhanced versions, named as TFS-grep and TFS-grep* to make the comparison. The experimental setup is presented in Appendix E. We use precision, recall and F1-score to measure the retrieval

Table 4: Comparison between LSFS and methods in the traditional file system (TFS) in retrieving files by keywords that match names and content of files.

| Metric | # files | TFS search window | TFS-grep | TFS-grep* | **LSFS** |
|---|---|---|---|---|---|
| Precision | 10 | 0.708 | 0.389 | 1.000 | 0.950 |
| | 20 | 0.724 | 0.396 | 1.000 | 0.870 |
| | 40 | 0.691 | 0.403 | 1.000 | 0.863 |
| Recall | 10 | 1.000 | 0.416 | 1.000 | 0.833 |
| | 20 | 1.000 | 0.292 | 1.000 | 0.933 |
| | 40 | 1.000 | 0.306 | 1.000 | 0.960 |
| F1-score | 10 | 0.829 | 0.402 | 1.000 | 0.891 |
| | 20 | 0.840 | 0.337 | 1.000 | 0.900 |
| | 40 | 0.817 | 0.348 | 1.000 | 0.909 |

performance. From the results presented in Table 4, we can find that LSFS outperforms TFS search window and TFS-grep, only second to the TFS-grep*. We find that the built-in retrieval tool in the TFS (e.g., the system search window) can not generate stable retrieval results, although it has a higher recall. Due to the fuzzy search feature in the built-in search window, it can easily retrieve inaccurate results for which the retrieved file content can only match part of the keywords. For example, if we search for an article written by *John Smith*, any other article with a name of *John* can be returned as a search result, thus the results may often include many irrelevant results, which complicates the process for users to filter through them. The TFS-grep* command, although achieving perfect results, still has several limitations. First, the commands can be too difficult to construct, especially when the file retrieval queries have multiple conditions. For instance, when a user requests to retrieve all files containing the keywords A and B, the command would be as follows: *find /path -type f -exec grep -l "keyword1" \; -exec grep -l "keyword2" \;*. Second, since the grep command itself does not support retrieval of binary files, it is necessary to manually adjust the format of each file, which is time-consuming and greatly reduces the efficiency of the retrieval process for users. Our LSFS can retrieve all types of text files, from plain text to binary files, while maintaining high precision and recall. The LSFS read operation is capable of processing both plain text and binary text, converting them into the vector database of system. This enables seamless retrieval operations across various types of files. Furthermore, LSFS allows users to describe their retrieval tasks in natural language, eliminating the need to write complex commands. The reults of performance of file sharing function is in Appendix H.2. It shows that our system can generate sharable files more efficiently.

## 6    CONCLUSIONS

In this paper, we present an LLM-based semantic file system (LSFS), which offers advancement over traditional file systems by enabling files to be stored and managed based on their semantic information. This enhancement improves the ability of system to comprehend and utilize the semantics embedded in file contents. Additionally, we introduce a series of reusable semantic syscalls and a framework for mapping natural language into LSFS parameters. These innovations provide a foundation for future research and development in the area of semantic file management systems.

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

APPENDIX

**Roadmap:** Section §A introduces the detailed implementations of LSFS sycalls. Section §B introduces the detailed implementations of LSFS APIs. Section §C shows concrete prompt examples to execute different APIs. Section §D shows the details of semantic file retrieval and keyword-based file retrieval. Section §I shows the results of a case study of comparing methods using LSFS and without using LSFS in semantic file retrieval task. Section §K presents analysis of file sharing tasks with pseudo-code examples of the file sharing process.

## A    THE IMPLEMENTATION DETAILS OF SYSCALL

**`create_or_get_file()`** This syscall integrates multiple traditional file system functions such as creating files, reading files, and opening files, enabling different operations based on the parameters provided. The function accepts four parameters: the LSFS path, the target directory name, the target filename, and the import file. The first two parameters are positional, while the latter two are default parameters. When all four parameters are provided, if the target file does not exist within LSFS, the system will create an imported file using the specified target directory name, target filename, and the import file. The import file can be supplied as a string or a file path, and our system supports various text file formats, including *PDF*, *DOCX*, *TXT* and so on. If the target filename is not passed, the syscall returns a list of files within the target directory. If the content of the import file is not provided, the syscall will return the target file, allowing access to its content, metadata, embedding vector, and other associated information.

**`add_()`** This syscall facilitates appending content to a file by accepting four positional parameters: the LSFS path, the target directory name, the target filename, and the content of the import file. The import file content can be provided either as a string or a text file in various formats. When all four parameters are supplied, the syscall appends the specified content to the designated file within the system.

**`overwrite()`** This syscall implements the overwriting of the file contents. The passed Parameters are also LSFS path, target directory name, target filename, import file, all of which are Positional Parameters. When passed in, LSFS will overwrite everything in the source file with the contents of the imported file.

**`del_()`** This syscall performs the deletion of files and directories by accepting four parameters: the LSFS path, the target directory name, the target filename, and the key text. The first two parameters are positional, while the last two are default parameters. If neither of the last two parameters is provided, the syscall raises an error, indicating that at least one must be passed. When the target filename is provided, LSFS deletes the specified file. If the key text is provided instead, the system searches for files containing the key text within the target directory and deletes them if found. Additionally, once all files within a directory are deleted, LSFS will automatically remove the directory.

**`keywords_retrieve()`** This syscall implements keyword search functionality of LSFS, retrieving all files within a specified directory that contain a given keyword. The arguments passed include the LSFS path, directory name, keyword, and matching condition. The LSFS path and keyword are Positional Parameters, while the directory name and condition are Default Parameters. If a directory name is provided, the syscall retrieves files within that directory that match the keyword; otherwise, it searches across the entire system. To match multiple keywords, the matching condition must be passed, specifying the relationship between keywords, such as *and* or *or*. The search results return a list of file names and a list of file contents.

**`semantic_retrieve()`** This syscall implements the semantic similarity search function within the LSFS, allowing retrieval of files that semantically match a given query within a specified directory. The parameters for this function include the LSFS path, the target directory, the search keyword, and the number of results to return. The LSFS path and search keyword are Positional Parameters, while the target directory and number of results are Default Parameters. Similar to keyword-based retrieval, the search directory will be determined based on whether a target directory is provided. The number of results to return dictates how many of the top-scoring matches are retrieved. Our semantic retrieval leverages the *LlamaIndex* framework. During file creation, a *LlamaIndex* vector store is generated alongside the index, enabling more intelligent and efficient data retrieval. This setup ensures that

semantic queries can return highly relevant results with improved accuracy. The search results return a list of file names and a list of file contents.

**create()** This syscall function facilitates batch directory creation and bulk file reading. It accepts the LSFS path, directory name, and the import file path as Positional Parameters. This function can read multiple files at once and store them in the specified target directory. If the filenames are not explicitly provided, they will default to the original filenames from the filesystem.

**lock_file() / unlock_file()** These syscalls handle the locking and unlocking of files within the LSFS, allowing files to be placed in read-only mode to prevent modification. Both syscalls accept the LSFS pathname, directory name, and filename as parameters. Upon execution, the lock_file() syscall updates the file's metadata to reflect a read-only state, effectively restricting any modifications. Conversely, the unlock_file() syscall modifies the metadata to restore read and write permissions, enabling the file to be edited again. These operations provide granular control over file access and modification rights.

**group_keywords()** This syscall groups all files containing a common keyword and creates a new directory for them. The parameters passed are the LSFS name, the keyword, the name of the new directory, the target directory, and the search criteria. Among these, the LSFS name, keyword, and new directory name are Positional Parameters, while the target directory and search criteria are Default Parameters. The syscall first performs a search using the keyword through an atomic syscall to identify all matching files. It then uses these files to create the specified new directory, facilitating easier file management and organization.

**group_semantic()** This syscall facilitates the organization of files by grouping all those containing a common keyword into a new directory. It takes the following parameters: the LSFS path, the keyword, the new directory name, the target directory, and the search criteria. Here, the LSFS path, keyword, and new directory name are Positional Parameters, while the target directory and search criteria are Default Parameters. The syscall first performs a keyword search through an atomic syscall to identify all files that match the keyword. It then creates a new directory with the specified name and moves the identified files into this directory. This functionality streamlines file management and enhances organizational efficiency.

**integrated_retrieve()** This syscall is designed for composite searches, combining both semantic and keyword search functionalities. The parameters are distributed as follows: the LSFS path, keyword, new directory name, and search criteria are Positional Parameters, while the target directory and additional search conditions are Default Parameters. The syscall first invokes the keyword grouping function to retrieve all files associated with the compound keyword and organizes them into a new directory. It then performs a semantic search within this directory, ultimately returning the filenames and contents of the files that match the search criteria. This integrated approach allows for more comprehensive and flexible search capabilities.

**file_join()** This syscall facilitates the connection of two files, with the parameters including the LSFS path, the directory name and filename of both files, and the connection conditions. The Default Parameters are the destination directory for *file 2* and the connection condition, while the LSFS path, directory names, and filenames are Positional Parameters. If the destination directory for *file 2* does not exist, the two files will be placed in the same directory. If the destination directory exists, the files will be placed in their respective directories. If the join condition is set to *new*, the syscall will preserve the original files, concatenate their text contents, and create a new file in the destination directory of *file 1*. The new name of file will be a combination of the two target filenames. If the join condition is not *new*, the contents of *file 2* will be appended directly to the contents of *file 1*, and *file 2* will then be deleted.

## B  THE IMPLEMENTATION DETAILS OF LSFS API

**Retrieve-Summary API.**    This API retrieves files based on user-specified conditions and provides concise summaries. Unlike traditional systems, it offers both keyword and semantic search, along with LLM-powered content summarization. The API supports three retrieval methods: keyword, semantic, and integrated, which are built on top of the keywords_retrieve(), semantic_retrieve(). and integrated_retrieve() syscalls. In this API, an interaction interface is also provided

for users to refine the results by excluding irrelevant files, which are then passed to LLMs for summarization.

**Change-Summary API.** This API is used to modify the contents of a file and compare them before and after to summarize the changes. At the same time, the Supervisor module is introduced to monitor the file changes in the traditional file system. Unlike traditional file systems that require tedious operating system commands, this API allows users to locate target files using natural language and automatically generate a summary of file changes through LLMs integration. The API is implemented by leveraging the supervisor and the `overwrite()` and `del_()` syscalls. In the change-summary API, when the target file is updated with the content of the source file, it will generate a summary of the modifications. Meanwhile, the filename is used as the key in the version recorder, while the metadata and contents of the file are stored as the corresponding values for the version control use.

**Rollback API.** This API is designed to achieve version restoration by utilizing the version recorder from the change-summary API, making version rollbacks more manageable. In this API, the `overwrite()` and `create_or_get_file()` syscalls are employed. Traditional file systems like ZFS, BtrFS, and NTFS offer rollback capabilities, but they primarily rely on snapshots, where the system captures the state of files at a specific point in time and restores them to that version. Our rollback API in the LSFS introduces two rollback methods for greater flexibility and ease of use. The first is time-based rollback, where the LLMs parses the rollback time from the input of user and reverts the file to the corresponding version. The second is version-based rollback, allowing users to specify in the prompt how many versions backward they wish to revert. This dual approach makes it easier for users to rollback to the target version of files.

**Link API.** This API is designed to enable the creation of shareable file links. In traditional file systems, links can only be generated for local access, limiting collaboration. However, with the *link API* of LSFS, shareable links can be generated for broader accessibility. Specifically, the API leverages cloud database, e.g., Google Drive, to upload files and generate the shareable link. Additionally, validity period can be passed as an argument for this API, once the period expires, the *link API* automatically revokes the link and terminates access for secure and time-bound file sharing.

## C  THE INSTRUCTION EXAMPLES OF EXECUTING API

This section introduces some instruction examples of executing LSFS APIs, which can be seen at Table 5.

Table 5: Some examples of instruction of API in Section 4.3. For every API, we provide different instructions in different task condition. The instruction of retrieve-summary API is in Appendix D.

| API Type | Instruction |
|---|---|
| **Change-Summary API** | **LSFS Input**: 
 w/ directory: Change the content of /xxxx/xxxx.txt to old-file under llm-directory. 

 w/o directory: Modify /xxxx/xxxx.txt to contain change-file. 

 **LLM Input**: At current step, you need to summary differences between the two contents, the content before the update is [*old file*], the content after the update is [*new file*] |
| **Rollback API** | **LSFS Input**: 
 By date: Revert the file named syntax to its version from 2023-6-15. 

 By version number: Rollback the cnn file to the state it was in 3 versions ago. |
| **Link API** | **LSFS Input**: 
 w/ period of validity: Provide a link for llm-base that will be active for 3 months. 

 w/o period of validity: Generate a link for system-architecture. |

## D    TASK DETAILS OF KEYWORD-BASED AND SEMANTIC RETRIEVAL.

In this section, Table 6 and Table 7 presents instruction with or without LSFS using LLM in semantic-based retrieval, and keyword-based retrieval (single-condition and multi-condition), respectively.

Table 6: The example of instruction of semantic-based retrieval with single-condition and multi-condition in LSFS and in LLM without LSFS.

| Task | Task Example | Method | Instruction |
|---|---|---|---|
| **Semantic-based Retrieval** | Locate the 3 papers showing the highest correlation with reinforcement learning in LLM-training. | **LLM w/o LSFS** | **Fixed prompt**: In the next step, you need to accept and remember the paper, but do not generate any outputs. Until you are told to output something.

**Each input**: The paper is [*content*].

**After every five entries**: Now you can to output the answer. You need to find [*retrieve number*] papers which most relate to [*retrieve condition*] from previous record and summary them respectively.

**Final input**: Now you can to output the answer. You need to choose from memory cache to find the [*retrieve number*] papers that is most relevant to [*retrieve condition*] |
| | | **LSFS** | **LSFS input**: Locate the 3 papers showing the highest correlation with reinforcement learning in LLM-training.

**LLM input**: You need to summary the content. The content is [*file content*] |

## E    BASELINES IN KEYWORD-BASED FILE RETRIEVAL AND FILE SHARING

The baselines to compare with LSFS in keyword-based file retrieval are as below:

- **TFS search window**: We use the default search window in the file of computer folder to retrieve files (i.e., Spotlight in MacOS) which supports retrieving by keywords in the file content.
- **TFS-grep**: We use Python program to convert the binary file to a plain text file and then perform *grep* operation on the converted plain text file.
- **TFS-grep***: After converting binary files into plain text, issues such as missing spaces, incorrect line breaks, and formatting errors may arise. In the *TFS-grep** process, we correct the format of the converted files and then run the *grep* operation on the properly formatted versions.

The baselines to compare with LSFS in file sharing are as below:

- **Gemini-1.5-flash**: We use Gemini-1.5-flash as the LLM backbone to write the code that generates the link for the target file, and then use the Python compiler to check the validity of the code.
- **GPT-4o-mini**: We employ GPT-4o-mini as the LLM backbone to generate code for creating links of the target file, followed by using the Python compiler to verify the validity of code.

Table 7: The example of instruction of keyword-based retrieval with single-condition and multi-condition in LSFS and in LLM without LSFS.

| Task | Task Example | Method | Instruction |
|---|---|---|---|
| **Keyword-based Retrieval(Single-Condition)** | Find papers in the computer-vision category authored by Emily Zhang. | **LLM w/o LSFS** | At current step, you need to judge if the input paper satisfy [*retrieve condition*]. If yes, you should summarize the paper, if no you do not need to output anything. The paper is [*file content*]. |
| | | **LSFS** | **LSFS input**: Find papers in the computer-vision category authored by Emily Zhang. 

 **LLM input**: You need to summary the content. The content is [*file content*] |
| **Keyword-based Retrieval(Multi-Condition)** | Find papers from either Cambridge University or Columbia University. | **LLM w/o LSFS** | At current step, you need to judge if the input paper satisfy [*retrieve condition*]. If yes, you should summarize the paper, if no you do not need to output anything. The paper is [*file content*] |
| | | **LSFS** | **LSFS input**: Find papers from either Cambridge University or Columbia University. 

 **LLM input**: You need to summary the content. The content is [*file content*]. |

- **AutoGPT**: We create a CoderGPT agent by initializing GPT as an expert on coding, which is used to generate the code.

- **Code Interpreter**: We use the Code Interpreter module of the OpenAI web client to generate relevant code and subsequently check the validity of the code.

## F  COMPARISON BETWEEN LSFS AND OPERATING SYSTEM

LSFS simplifies operations by reducing the time users spend learning and inputting commands. To provide a clearer understanding of the time costs, we present a comprehensive time analysis below. Traditional command execution typically follows this workflow: learn command parameters (via ChatGPT or search engines) → locate appropriate file paths through OS lookup mechanisms →input the complete command → finally obtain results. In contrast, LSFS streamlines this process into a simpler workflow: input natural language commands with file names or semantic keywords → confirm LSFS suggested file operations and obtain results. The result in the 8.

In order to quantitatively analyze the time difference between traditional file systems and LSFS, we adopt FS commands and LSFS commands to achieve different file operations shown below. 10 Ph.D. student are invited perform file operations through using Linux commands in traditional file systems and using natural language commands in LSFS, respectively. The result in the 9

The breakdown of time in different steps for the students to operate files correctly is collected and the average time in each step is calculated as below. Although the time for the Learn Command step decreases to zero for skilled users, the sum of the rest of the operations performed by traditional filesystems is still greater than the sum of LSFS.

Table 8: Operating Complexity of LSFS and Traditional File System.

| Operation | Traditional FS | LSFS |
|---|---|---|
| Keyword-retrieve | find /path -type f -exec grep -l "keyword1" ; -exec grep -l "keyword2" | Find the file contains 'keyword1' and 'keyword2' |
| Rollback | btrfs subvolume snapshot /path/to/directory /path/to/snapshot btrfs subvolume delete /path/to/directory btrfs subvolume snapshot /path/to/snapshot /path/to/directory | Rollback the 'filename' to the version in 'date' |
| Group By | mkdir -p /path/to/new-folder 
 find /path/to/search-folder -type f -exec grep -l "keywords" ; -exec mv /path/to/new-folder/ ; | *group-keywords* with input: search-folder, keywords, new-folder |
| Join | cat /path/to/file1.txt /path/to/file2.txt > /path/to/new-file.txt | file-join syscall with input: file1, file2, new-file |
| Link | ln -s /home/user/file-name /home/user/shortcut/data-link | Create a link for file-name |

Table 9: Time comsuming of LSFS and Traditional File System in each step

| Input Command | LSFS Parser | Task Execution | Total | - |
|---|---|---|---|---|
| 11.43(s) | 4.21(s) | 11.95(s) | 27.59(s) | - |
| Learn Command | Find Path | Input Command | Task Execution | Total |
| 153.61(s) | 28.23(s) | 30.30(s) | 0.02(s) | 212.16(s) |

# G  SECURITY MECHANISM

We designed some security mechanisms: ❶ We added a process lock to LSFS to prevent consistent reads and writes to the same file. ❷ We design a user confirmation step: When a user makes a change to a file, the user will be asked to confirm the changed object twice. ❸ We designed rollback operations: if the user makes a wrong change to the file, they can roll back to the correct version. For first and third mechanisms, the file operation reliability can be improved as long as these two mechanisms are enabled. For the second mechanism, we conducted a quantitative evaluation of two aspects: the probability of file misoperation and the proportion of risky operations. We compared these metrics with and without the confirmation mechanism enabled. In the current experiment, we used the retrieval function to locate target files for operations. Table. 10 below shows the probability of retrieval errors with and without the confirmation step. We can see that after adding user confirmation, the retrieval error is reduced to 0%. Additionally, we evaluated the proportion of potentially dangerous

Table 10: Retrieval errors with and without the confirmation step

| # files | Without User Confirmation | With User Confirmation |
|---|---|---|
| 10 | 13% | 0% |
| 20 | 16.7% | 0% |
| 40 | 15.8% | 0% |
| 120 | 14.8% | 0% |

operations executed (e.g., write, update, or delete) across all file management APIs. The percentage without user confirmation was 36.8%. Some dropped to 0%.The results below demonstrate that the confirmation mechanism in LSFS effectively prevents unintended dangerous operations.

# H  FURTHER RESULTS OF LSFS PARSER

## H.1  SUCCESS RATE OF LSFS

Since in Sec. 5.1, the extraction success rate in our LSFS Parser is not 100%, We conduct the case study of the incorrect results and find that LSFS parser sometimes performs bad in parsing complex commands due to the capability and inherited randomness of the LLM. To further improve the reliability of our system, we make the following enhancements. We add the failure case to the prompt like the result of ***failure case*** is wrong, you should refer to the case and regenerate it. Then let LSFS parser generate the answer again, the experiment results as follow:

Table 11: Parser accuracy at Second Parsing Success Rate for the four models

| Model | Operation | First Parsing Success Rate | Second Parsing Success Rate |
|---|---|---|---|
| Gemini-1.5 | Retrieve-Summary API | 100% | - |
| | Change-Summary API | 96.7% | 100% |
| | Link API | 100% | - |
| | Rollback API | 83.3% | 100% |
| GPT-4o-mini | Retrieve-Summary API | 91.3% | 100% |
| | Change-Summary API | 100% | - |
| | Link API | 100% | - |
| | Rollback API | 100% | - |
| Qwen2:7b | Retrieve-Summary API | 86.7% | 100% |
| | Change-Summary API | 100% | - |
| | Link API | 100% | - |
| | Rollback API | 83.3% | 100% |
| Gemma:2b | Retrieve-Summary API | 76.7% | 85.7% |
| | Change-Summary API | 96.7% | 100% |
| | Link API | 91.3% | 100% |
| | Rollback API | 100% | - |

In the Table. 11. After a second parsing using the use cases that were incorrectly parsed the first time, we can see that all LLM backbone achieved 100% accuracy on each task except Gemma-2, which did not achieve 100% accuracy on the Retrieve-Summary API. This means that our parser can parse the task correctly at most twice on most tasks and llm backbone, so we consider the parser to be useful for mapping work. For data collection, we invited 10 Ph.D. students to write preliminary instructions for related tasks. Then the results are used for testing, and the instruction information is fine-tuned according to the test results, and finally the instruction with the highest score is selected.

## H.2 Performance Analysis in File Sharing

For the file sharing task, we evaluate whether a system can output a shareable link with an expiration time according to the prompts.

Specifically, we compare LSFS with four different baselines which details are in Appendix E.

Table 12: Comparison between LSFS and other LLM-leveraged methods in File Sharing.

| Method | Success rate of generating sharable links (#20) | | | |
|---|---|---|---|---|
| | Code Generation Rate | Link Generation Rate | Link Validness Rate | Final Success Rate |
| Gemini-1.5-flash | 65% | 45% | 45% | 10% |
| GPT-4o-mini | 60% | 35% | 30% | 5% |
| AutoGPT | 50% | 45% | 15% | 5% |
| Code Interpreter | 100% | 75% | 65% | 0% |
| **LSFS** | **100%** | **100%** | **100%** | **100%** |

In the experiments, four key metrics are used to evaluate the effectiveness of whether the system can successfully fulfill the file sharing task: whether the LLM generates code, the correctness of the generated code, the effectiveness of the generated links, and whether the links are actually shareable. We evaluate with 20 file sharing task prompts for all the methods. The results show that although all methods even vanilla LLMs can successfully generate code, they do not consistently generate valid links. In many cases, these links are local rather than shareable links, and only a small fraction of the links for files are shareable. In contrast, our LSFS system achieves 100% link generation success rate, showing strong task fulfillment ability on the file sharing task.

## I  A Case study of Semantic File Retrieval

We conduct a case study using the example prompt "*Please search for the two papers most related to LLMs Uncertainty from folder named example*" to better illustrate the retrieval results, which is shown in the Figure 7.

For the method without using LSFS, the answer to the intermediate result is "*GNN-RAG: Graph Neural Retrieval for Large Language Model Reasoning···,*" which fails to identify a target paper

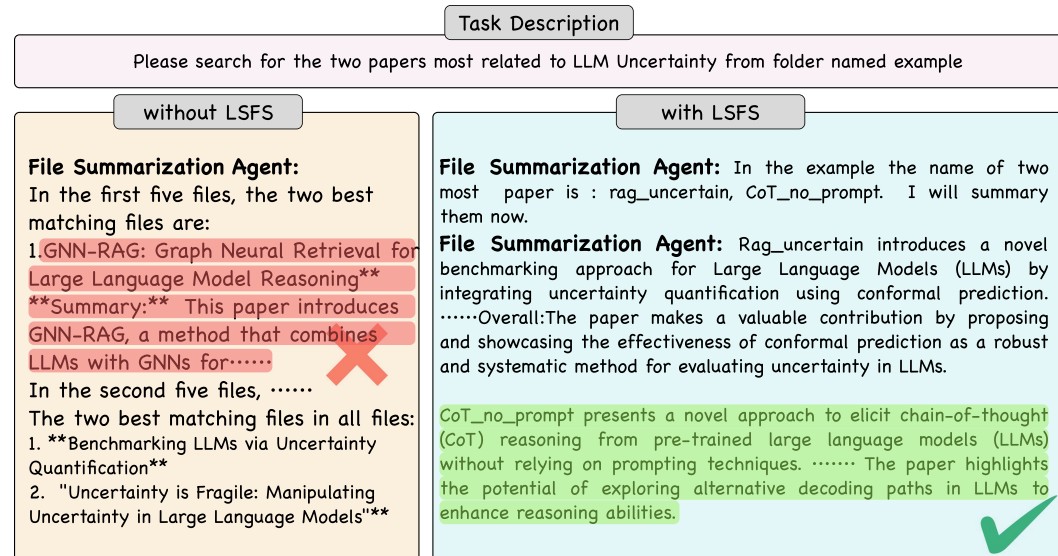

Figure 7: A case study of comparing semantic file retrieval between LLM-leveraged methods without using LSFS and using LSFS.

that is relevant to LLMs Uncertainty due to the long-context that makes it difficult for LLM to understand and retrieve the correct information. Furthermore, the input length limitation of LLM necessitates a batch-processing strategy for file input, which can result in selecting the "best of the worst" candidates. This may lead to inaccurate intermediate results that ultimately impact the final output. In contrast, LSFS avoids this issue by evaluating all files holistically, scoring and sorting the final results without intermediate outputs, thus bypassing the constraints of limited search scope. Experimental results show that LSFS consistently delivers more accurate results.

## J  FUTURE WORK

Looking ahead, various future directions can be further explored. **1) Multi-modal and multi-extension file management:** Currently, LSFS primarily supports operations on text files. Semantic operations for understanding and managing non-text files, such as *XLSX*, *JPG*, *MP3*, and *MP4* can be further optimized. **2) Security and privacy enhancements:** Data encryption techniques can be explored to secure data interactions and transmissions between LSFS and LLMs, ensuring that file privacy remains protected at all stages of processing and communication. **3) Optimized retrieval strategies:** Retrieval methods can be further optimized by integrating more advanced and precise algorithms, enhancing the overall accuracy and effectiveness of retrieval performance of LSFS. **4) More instantiated APIs and syscalls:** While this paper focuses on the design of the most essential and commonly used syscalls within LSFS, more functional APIs and syscalls can be explored to fulfill user's envolving requirements. We believe explorations on these directions can help to expand the functionalities of LSFS for a more intelligent and user-friendly operating system.

## K  CODE EXAMPLE OF FILE SHARING

In this section, we analyze the failure cases of file sharing in baselines and use some pseudo-code examples to show the file sharing process in baselines and LSFS, respectively. For all the methods, we set the following input: *You are good at writing code, please write code to generate shared links for the file 'path'* as the system prompt for backbone LLMs.

### K.1  THE CODE CANNOT GENERATE LINK

In the experiment, the code generated by LLMs may not produce the correct link or link address. For instance, even after successfully installing the required file packages, the following code block

demonstrates that the generated link does not direct to the intended target file. The pseudo-code is in Algorithm 1.

---

**Algorithm 1** Pseudo-code of K.1.

---

```
app = Flask(__name__)
# Path to the PDF file
pdf_file_path = '/xxxx/xxx.pdf'
@app.route('/download')
def download_file():
    return send_file(pdf_file_path, as_attachment=True)
if __name__ == '__main__':
    app.run(debug=True)
```

---

## K.2    THE CODE CAN ONLY GENERATE LOCAL LINK

In most cases, the generated code will produce links to the corresponding files. The code block typically appears as shown below; however, the links generated by this code are limited to local access and do not provide shareable links for external users. The pseudo-code is in Alg. 2

---

**Algorithm 2** Pseudo-code of K.2.

---

```
# Define the file path
file_path = Path('/xxxx/xxx.pdf')

# Check if the file exists
if not file_path.exists():
    raise FileNotFoundError(f"The file {file_path} does not exist.")

# Copy the file to a shareable directory (e.g., a public folder)
shareable_directory = Path('/mnt/data/shareable_files')
shareable_directory.mkdir(parents=True, exist_ok=True)

# Define the new path in the shareable directory
shared_file_path = shareable_directory / file_path.name

# Copy the file to the shareable directory
shutil.copy(file_path, shared_file_path)

# Generate a shareable link (assuming a file server is available at /mnt/
    data)
shareable_link = f"http://file-server-url/shareable_files/{file_path.name}"
print(shareable_link)
```

---

## K.3    THE CODE CAN GENERATE SHAREABLE LINK

In our experiments, the code generated by LLMs can occasionally produce a shareable link. However, generating such a link often involves complex configuration steps. For instance, users need to authorize the Dropbox app, obtain an access token, and perform other setup tasks, as illustrated in the following code block. Moreover, due to the variety of platforms for generating shareable links, LLMs may switch between different platforms with each code generation, leading to considerable user configuration time. The Steps and Pseudo-code in Alg. 3

---

**Algorithm 3** Procedures of K.3.

---

1: Install the DropBox SDK.
2: Once logged in, use the APP Console to create a new app and select the appropriate permissions.
3: Configure application permissions on demand.
4: Create an access token using OAuth2.
5: File generator:

       • Import the private token of OAuth2: accesstoken = ′Your token′
       • Create a dropbox client: dpclient = Dropbox(accesstoken)
       • Import file path: path = ′ xxx/xxxx.pdf ′
       • Use dropbox to create a shared link: link = dpclient.share(path)

6: Get the link

---

In contrast, our *Link API* simplifies this process: users only need to provide Google Drive credentials, and they can effortlessly generate shareable links without the need for extensive configuration.

