# OpenReview forum: "From Commands to Prompts: LLM-based Semantic File System for AIOS"
_ICLR.cc/2025/Conference — ICLR 2025 Poster_

### Official Review · Reviewer_aTqT · 2024-10-20

**Soundness:** 3
**Presentation:** 3
**Contribution:** 3
**Rating:** 8
**Confidence:** 3

**Summary:**

The paper represents a problem in the current scenario of semantic file matching algorithms; currently, we use the traditional way of semantic matching algorithms based on file name, size, and timestamps. This involved remembering syntax and filenames. This fails in scenarios where two files have similar text; here its hard to distinguish files based on pure string matching. The paper introduces LLM with traditional file systems to do LLM based semantic file management.

LSFS extracts semantic features from the file content and generates corresponding embedding vectors. LSFS incorporates semantic information into its file operations. In Linux, if we need to change a file, i.e., replace a file with another, we need to remember the path, but with LSFS  the users don't need to remember the file name and can talk to LLM to make the changes for them. They have introduced a LLM-based Semantic File System (LSFS), an LSFS parser, and safety insurance mechanisms to the traditional file matching algorithms. The paper has done a great job at explaining the traditional way and modifications done with NLP. They have elaborately explained the API changes they have made over traditional architecture and given diagrams to explain the architecture. Also, they have demonstrated how components of LSFS interact with each other to achieve different functionalities.

Evaluations are carried out based on success, performance, and performance on non-semantic based tasks like file sharing over sample data/files.

**Strengths:**

The paper touches on an existing problem that exists in the day-to-day lives of developers and Mac OS users of remembering the file names and directory where the files are present and need modification. There is no way to solve this problem at the present. Even while using LLMs, sometimes developers have to hard code the file path for retrieval. LLM based file retrival system is new and useful for anyone who is fed up of the traditional based systems.They did pretty well work on describing the APIs to be used in the new framework and a commendable job in comparing the APIs to the traditional ones. The quality of the paper was good and the presentation with diagrams were very useful to get the context of the paper.
The architecture of the new framework was explained in detail and they have done a good job in explaining how each component in the architecture is integrated with LLMs. Evaluations are carried out based on success, performance, and performance on non-semantic based tasks like file sharing over sample data/files and are pretty easy to follow.

**Weaknesses:**

The paper could have used an example to walkthrough the implementation. Each component description could have been presented with design diagrams or a flowchart that is easy to understand; visual representation always helps! More evaluations to prove their architecture is better than the traditional ones based on performance, latency, operational burden, and cost. The paper didn't touch on any security concerns while using the LLMS. Are there guardrails in place to restrict the LLMs to scanning through the personal data? One more thing the paper lacked was elaborating on the use cases where this architecture can be used.

**Questions:**

1) Are there guardrails in place to restrict the LLM to not scan through personal data?
2) How much cost are we saving with the new architecture?
3) Are there any security concerns for using this architecture?
4) How much of an operational overhead is this architecture based on traditional architecture ?
5) What are the other use cases of this architecture in real life scenarios?
6) This seems like an ongoing problem that needs to be resolved; are there any similar existing architectures? Have you looked at those papers?
7) Is there an Andon Cord mechanism to stop the LLM to give out hallucinations and wonky results to the user?
8) While scanning through the files, is the data saved in memory? Does the data contain PII information (ppersonal information about the user)?

---

> ### Author Response · Authors · 2024-11-20
> **Author Response(1/3)**
>
> Thank you very much for taking your valuable time to review our paper and provide us with many constructive suggestions. Below are our detailed responses to your concerns:
> > ***W1: The paper could have used an example to walk through the implementation. Each component description could have been presented with design diagrams or a flowchart that is easy to understand; visual representation always helps!***
>
> Thank you for raising this question. A finer-grained example to walk through the components has been added in the Fig.1. in the revised version.
>
> > ***W2: More evaluations to prove their architecture is better than the traditional ones based on performance, latency, operational burden, and cost.***
>
> > ***Q2: How much cost are we saving with the new architecture?***
>
> >  ***Q4: How much of an operational overhead is this architecture based on traditional architecture?***
>
> Compared to traditional file systems, LSFS offers enhanced functionality, such as grouping and semantic retrieval, and performs better in keyword-based retrieval tasks. However, LSFS introduces additional overhead, including LLM inference and database operations, leading to longer execution times compared to traditional systems. Despite this, LSFS simplifies operations by reducing the time users spend learning and inputting commands. To provide a clearer understanding of the time costs, we present a comprehensive time analysis below.
>
> | operation  |  Traditional FS |  LSFS |
> |---------|---------|---------|
> | Keyword_retrieve   |  · find /path -type f -exec grep -l ”keyword1” \; -exec grep -l ”keyword2” \;  |  Find the file contains 'keyword1' and 'keyword2'  |
> |  Rollback  | · btrfs subvolume snapshot /path/to/directory /path/to/snapshot <br> · btrfs subvolume delete /path/to/directory <br> · btrfs subvolume snapshot /path/to/snapshot /path/to/directory |  Rollback the 'filename' to the version in 'date' |
> | Group by | · mkdir -p /path/to/new_folder <br> · find /path/to/search_folder -type f -exec grep -l "keywords" {} \; -exec mv {} /path/to/new_folder/ \; | *group_kewords* with input : search_folder, keywords, new_folder | Change   | cat /path/to/source_file.txt | tee -a /path/to/destination_file.txt | Change "source_file" by '/path/to/destination_file.txt' |
> | Join | cat /path/to/file1.txt /path/to/file2.txt > /path/to/new_file.txt | *file_join* syscall with input: file1, file2, new_file|
> | Link     | ln -s /home/user/file_name /home/user/shortcut/data_link  |    Create a link for file_name     |
>
> In order to quantitatively analyze the time difference between traditional file systems and LSFS, we invited 10 Ph.D. to execute the above commands with Linux commands in traditional file systems and natural language commands in LSFS, and calculated the time consumption of each part according to our above time division. The results are as follows:
>
> **LSFS**
> | Input Command  |  LSFS Parser |  Task Execution | Total|
> | ---------|---------|---------|---------|
> |  11.43s  |   4.21s    |   11.95s | 27.59s |
>
> **Traditional FS**
> | Learn Command  |  Find Path | Input Command  |Task Execution | Total|
> | ---------|---------|---------|---------|---------|
> |  153.61s  |   28.23s    |   30.30s | 0.02s |  212.16s   |
>
> > ***W3: The paper didn't touch on any security concerns while using the LLMS. Are there guardrails in place to restrict the LLMs to scanning through the personal data?***
>
> >  ***Q1: Are there guardrails in place to restrict the LLM to not scan through personal data?***
>
> In LSFS, our user filter is implemented to prevent LLMs from scanning to personal information. When LSFS retrieves an eligible file to perform subsequent tasks, it will first send the user confirmation, and if the user finds that the retrieval results contain personal information, he can cancel sending the file to LLM.

---

> > ### Comment · Reviewer_aTqT · 2024-11-22
> >
> > "In LSFS, our user filter is implemented to prevent LLMs from scanning to personal information. When LSFS retrieves an eligible file to perform subsequent tasks, it will first send the user confirmation, and if the user finds that the retrieval results contain personal information, he can cancel sending the file to LLM." - The guardrails should be there in place to avoid passing personal data , there should be guardrails for input and output from LLM. It should not be a manual process where user has to interviene  and cancel sending the file to LLM.

---

> > > ### Author Response · Authors · 2024-11-23
> > >
> > > Thank you for your recognition of our work. Guardrails are an important and worthwhile issue to explore, and we will aim to address this issue in future research.

---

> > > > ### Comment · Reviewer_aTqT · 2024-11-26
> > > >
> > > > Got it, thanks for taking it into consideration.

---

> ### Author Response · Authors · 2024-11-20
> **Author Response(2/3)**
>
> > ***W4: One more thing the paper lacked was elaborating on the use cases where this architecture can be used.***
>
> > ***Q5: What are the other use cases of this architecture in real life scenarios?***
>
> Semantic-based file systems are designed to cater to a broad range of usecases. For instance, as highlighted in [1], traditional file systems often impose a cumbersome approach to organizing documents, posing significant challenges for small and medium-sized enterprises (SMEs), public administration bodies, and individual users. LSFS addresses this gap by enabling more efficient organization and management of document content, reducing operational complexity for non-computer practitioners, including SMEs and public administration agencies. Furthermore, [2] identifies semantic file systems as a critical development trend, emphasizing their versatility and applicability across various domains in information technology (IT). Additionally, LSFS can fill a significant gap in current LLM-based multi-agent systems, as noted in [3], which often lack robust mechanisms for managing interaction records and background knowledge. By providing a more intuitive file management framework, LSFS benefits both end-users and system designers by enhancing file access and reducing administrative overhead.
>
> > ***Q3: Are there any security concerns for using this architecture?***
>
> Thank you for raising the question. We designed several mechanisms to address potential security concerns:
> 1. We added a process lock to LSFS to prevent consistent reads and writes to the same file
> 2. We design a user confirmation step: When a user makes a change to a file, the user will be asked to confirm the changed object twice
> 3. We designed rollback operations: if the user makes a wrong change to the file, they can roll back to the correct version
>
> For first and third mechanisms, the file operation reliability can be improved as long as these two mechanisms are enabled.
>
> For the second mechanism, we conducted a quantitative evaluation of two aspects: the probability of file misoperation and the proportion of risky operations. We compared these metrics with and without the confirmation mechanism enabled.
>
> In the current experiment, we used the retrieval function to locate target files for operations. The table below shows the probability of retrieval errors with and without the confirmation step:
>
> | Number of Files  | Without User Confirmation | With User Confirmation |
> |------------------|---------------------------|-------------------------|
> | 10               | 13%                       | 0%                     |
> | 20               | 16.7%                     | 0%                     |
> | 40               | 15.8%                     | 0%                     |
> | 120              | 14.8%                     | 0%                     |
>
> Additionally, we evaluated the proportion of potentially dangerous operations executed (e.g., write, update, or delete) across all file management APIs. The results below demonstrate that the confirmation mechanism in LSFS effectively prevents unintended dangerous operations:
>
> | Without User Confirmation | With User Confirmation |
> |---------------------------|-------------------------|
> | 36.8%                     | 0%                     |
>
> These results highlight that enabling the confirmation mechanism significantly enhances the safety and reliability of file operations in LSFS.
>
> > ***Q6: This seems like an ongoing problem that needs to be resolved; are there any similar existing architectures? Have you looked at those papers?***
>
> As highlighted in our related work, existing semantic file systems integrate semantics only into metadata and do not leverage large models to create a fully semantic file system.
>
> To the best of our knowledge, no work has systematically proposed semantic file management based on LLMs. Existing LLM-agent systems focus on enhancing agent functionalities through leveraging file content while neglecting the fundamental file management based on semantics. Agent frameworks with storage mechanism enabled, such as Autogen[4], Camel[5], and OS-Copilot[3], build user profiles and obtain knowledge from files. Knowledge in these systems typically stores agents' past interactions or acquired different files. However, agent systems built by these frameworks face limitations due to their reliance on traditional file management methods. One significant challenge is the need for developers to explicitly set up file paths for agents, which becomes increasingly cumbersome as the number of agents and agent-related files (e.g., task completion records and knowledge bases) grows. This limitation hinders scalability and efficiency in deploying multiple agents. Recognition of the limitation in existing agent systems inspires us to propose a semantic file system that supports the development of LLM-based agents, enabling more efficient and scalable semantic file management to support building of agents.

---

> > ### Comment · Reviewer_aTqT · 2024-11-22
> >
> > This clearly answers my questions. Thanks a lot for elaborating on it

---

> ### Author Response · Authors · 2024-11-20
> **Author Response(3/3)**
>
> > ***Q7: Is there an Andon Cord mechanism to stop the LLM to give out hallucinations and wonky results to the user?***
>
> First, our parser helps mitigate the hallucination problem by incorporating the fault-tolerance mechanism. If the LSFS parser incorrectly parses a user's command and fails to map it to the API, the failure case will be added into the prompt like *the result of {failure case} is wrong, you should refer to the case and regenerate it.* Then let LSFS parser generate the answer again. Secondly, our rollback mechanism provides a safeguard against potential LLM hallucinations during file operations. If the LLM erroneously modifies a file, the user can restore it to its previous version using the rollback feature.
>
>
> > ***Q8: While scanning through the files, is the data saved in memory? Does the data contain PII information (ppersonal information about the user)?***
>
> In the LSFS architecture, large language models (LLMs) serve as downstream processing tools. Users cannot add files to LSFS through LLMs, nor does LSFS provide interactive memory functionalities for LLMs. Consequently, files scanned or referenced during LLM interactions are not automatically added to the LSFS system's memory. Instead, the files present in LSFS are explicitly added by the user. Besides, to help protect the personal information of user, we provide the user confirmation mechanisms, which is detailed in our response to Q1.
>
> [1] D. Di Sarli, F. Geraci, “GFS: a Graph-based File System Enhanced with Semantic Features”, Proceedings of the 2017 International Conference on Information System and Data Mining, pp. 51-55, April 1-3, 2017
>
> [2] Mashwani, S.R. and Khusro, S. 2018. The Design and Development of a Semantic File System Ontology. Engineering, Technology & Applied Science Research. 8, 2 (Apr. 2018), 2827–2833. DOI:https://doi.org/10.48084/etasr.1898.
>
> [3] Zhiyong Wu, Chengcheng Han, Zichen Ding, Zhenmin Weng, Zhoumianze Liu, Shunyu Yao, Tao Yu, and Lingpeng Kong. Os-copilot: Towards generalist computer agents with self-improvement. arXiv preprint arXiv:2402.07456, 2024
>
> [4] Guohao Li, Hasan Hammoud, Hani Itani, Dmitrii Khizbullin, and Bernard Ghanem. Camel:Communicative agents for "mind" exploration of large language model society. Advances in Neural Information Processing Systems, 36, 2023.
>
> [5] Qingyun Wu, Gagan Bansal, Jieyu Zhang, Yiran Wu, Shaokun Zhang, Erkang Zhu, Beibin Li,Li Jiang, Xiaoyun Zhang, and Chi Wang. Autogen: Enabling next-gen llm applications via multi-agent conversation framework. arXiv preprint arXiv:2308.08155, 2023.

---

> > ### Comment · Reviewer_aTqT · 2024-11-22
> >
> > This answers my questions. Thanks!

---

### Official Review · Reviewer_a3yw · 2024-11-03

**Soundness:** 2
**Presentation:** 2
**Contribution:** 2
**Rating:** 3
**Confidence:** 4

**Summary:**

The authors propose, implement and describe an LLM-based semantic file system, where commands are replaced by prompts. They describe APIs and in several experiments compare how this filesystem is used and performs, compared to a conventional file system.

**Strengths:**

- very interesting and original setup
- includes interesting examples how to use the file system

**Weaknesses:**

- weak evaluation, based on examples and not on more extensive use cases and human evaluation, evaluation setup not described in detail / cannot be reproduced
- unclear for which users this could be helpful
- unclear how robust the system is when translating NLP input into actual actions
- unclear how the new API maps and extends conventional file handling APIs, and why setting up a new API set is superior to adding some APIs to a conventional file system

**Questions:**

1. Your systems seems to have advantages for searching and retrieving files based on keyword or semantic search. This could be implemented on top of a conventional file system, why implement a new API for that?
2. Is the accuracy of the LSFS parser of about 90% enough for meaningful work? That means 10% incorrect results. How did you collect the NLP commands for this evaluation?
3. How exactly did you perform the evaluation summarized in table 2?

---

> ### Author Response · Authors · 2024-11-20
> **Author Response(1/2)**
>
> We are deeply grateful for your valuable time and insightful feedback. Below are our detailed responses to your concerns:
>
> > ***W1: weak evaluation, based on examples and not on more extensive use cases and human evaluation, evaluation setup not described in detail / cannot be reproduced***
>
> We would like to further clarify our experiment setting as below.
> 1. In Experiment 5.1, we validate the effectiveness of our parser using four different LLM backbone. For each API, we created 30 use cases, each with a unique language structure. The test data is provided in the anonymous link.
> 2. In Experiment 5.2, we evaluated the effectiveness of LSFS in semantic retrieval. Since traditional file systems lack semantic retrieval capabilities, we used LLM alone as the baseline for file retrieval in our evaluation. Specifically, we measured the time and accuracy of executing a semantic file retrieval task by simply feeding file content into LLM to retrieve files and retrieving files using LSFS. Our target files contain various types of files such as ".txt,.pdf,.doc", etc. To make our experiments more extensive use cases, we supplemented with experiments under 120 files, and the results are as follows:
>
> | LLM-backbone |  Retrieval Accuracy w/o LSFS |  Retrieval Accuracy w/  LSFS | Retrieval Time w/o  LSFS | Retrieval Time w/  LSFS |
> |---------|---------|---------|---------|---------|
> | Gemini-1.5-flash|  35.2% | 92.9%(164%&#8593;)| 605.59s |48.08s(92.1%&#8595;) |
> | GPT-4o-mini| 63.8% | 92.9%(45.6%&#8593;) | 938.68s | 88.93s(90.5%&#8595;)|
>
>
> 3. In Experiment 5.2, in order to check the robustness of our rollback function when the number of files increases, we tested the rollback time under 5-40 versions respectively, and obtained the robustness of the rollback time.
>
> 4. In Experiment 5.3, we compared LSFS with traditional file retrieval methods based on Precision, Recall, and F1 score. The methods include:
>     - TFS-search-window: Uses the computer's search window (e.g., MacOS Spotlight) to retrieve both binary and plain text files.
>     - TFS-grep: Uses the Linux terminal command grep, which can only retrieve plain text files.
>     - TFS-grep*: An enhanced version of TFS-grep that first converts binary files into plain text before using grep to retrieve.
>
> 5. In the second part of Experiment 5.3, we compared LSFS and pure prompting to different LLMs as baselines to generate code for creating sharable file links.
>
> Finally, we would like to highlight that our code is provided through the anonymous link and **it's all reproducible**.
>
> > ***W2: unclear for which users this could be helpful***
>
> Semantic-based file systems are designed to cater to a broad range of users. For instance, as highlighted in [1], traditional file systems often impose a cumbersome approach to organizing documents, posing significant challenges for small and medium-sized enterprises (SMEs), public administration bodies, and individual users. LSFS addresses this gap by enabling more efficient organization and management of document content, reducing operational complexity for non-computer practitioners, including SMEs and public administration agencies. Furthermore, [2] identifies semantic file systems as a critical development trend, emphasizing their versatility and applicability across various domains in information technology (IT). Additionally, LSFS can fill a significant gap in current LLM-based multi-agent systems, as noted in [3], which often lack robust mechanisms for managing interaction records and background knowledge. By providing a more intuitive file management framework, LSFS benefits both end-users and system designers by enhancing file access and reducing administrative overhead.

---

> ### Author Response · Authors · 2024-11-20
> **Author Response(2/2)**
>
> >***W3: unclear how robust the system is when translating NLP input into actual actions.***
>
> > ***Q2: Is the accuracy of the LSFS parser of about 90% enough for meaningful work? That means 10% incorrect results. How did you collect the NLP commands for this evaluation?***
>
> We conduct the case study of the incorrect results and find that LSFS parser sometimes performs bad in parsing complex commands due to the  capability and inherited randomness of the LLM. To further improve the reliability of our system, we make the following enhancements. We add the failure case to the prompt like ***the result of {failure case} is wrong, you should refer to the case and regenerate it.*** Then let LSFS parser generate the answer again, the experiment results as follow:
>
> **Gemini-1.5**
> | Operation    | First Parsing Success Rate        | Second Parsing Success Rate |
> |--------------|-----------------------------------|-----------------------------|
> | Retrieve-Summary API| 100%| - |
> | Change-Summary API  | 96.7% | 100%|
> | Link API    | 100%  |  - |
> | Rollback API    | 83.3%  | 100% |
>
> **GPT-4o-mini**
> | Operation    | First Parsing Success Rate        | Second Parsing Success Rate |
> |--------------|-----------------------------------|-----------------------------|
> | Retrieve-Summary API| 91.3% | 100% |
> | Change-Summary API  | 100% |  -  |
> | Link API    | 100%  | -  |
> | Rollback API | 100%| -  |
>
> **Qwen2:7b**
> | Operation    | First Parsing Success Rate        | Second Parsing Success Rate |
> |--------------|-----------------------------------|-----------------------------|
> | Retrieve-Summary API|  86.7% | 100%  |
> | Change-Summary API  | 100%  | - |
> | Link API    |  100%  |  -  |
> | Rollback API    |  83.3% | 100%  |
>
> **Gemma:2b**
> | Operation    | First Parsing Success Rate        | Second Parsing Success Rate |
> |--------------|-----------------------------------|-----------------------------|
> | Retrieve-Summary API| 76.7% | 85.7%  |
> | Change-Summary API   | 96.7% | 100%|
> | Link API    | 91.3%      |  100%  |
> | Rollback API   | 100%| -  |
>
> After a second parsing using the use cases that were incorrectly parsed the first time, we can see that all LLM backbone achieved 100% accuracy on each task except Gemma-2, which did not achieve 100% accuracy on the Retrieve-Summary API. This means that our parser can parse the task correctly at most twice on most tasks and llm backbone, so we consider the parser to be useful for mapping work. For data collection, we invited 10 Ph.D. students to write preliminary instructions for related tasks. Then the results are used for testing, and the instruction information is fine-tuned according to the test results, and finally the instruction with the highest score is selected.
>
> > ***W4: unclear how the new API maps and extends conventional file handling APIs, and why setting up a new API set is superior to adding some APIs to a conventional file system.***
>
> > ***Q1:Your systems seems to have advantages for searching and retrieving files based on keyword or semantic search. This could be implemented on top of a conventional file system, why implement a new API for that?***
>
> Thank you for the recognition of the advantages of our LSFS. The building of APIs by simply combining system calls to achieve functionalities can not consider the semantic information of file in the file's lifecycle including creation, update, deletion. Therefore, we can not use them to achieve flexible semantically related APIs. Under this consideration, we design a new set of APIs to build the semantic index of file and support more flexible semantic file management operations.
>
>
> > ***Q3: How exactly did you perform the evaluation summarized in table 2?***
>
> In the "w/o LSFS" scenario, we input each text directly into the LLM sequentially, checking if it meets the target condition. If the condition is satisfied, the subsequent action is executed; otherwise, the text is skipped. In contrast, under the "w/ LSFS" setup, LSFS performs an initial search and prompts the user for confirmation. The corresponding file from the search results is then directly fed into the LLM for further processing. Additional details can be found at Appendix D.
>
> [1] D. Di Sarli, F. Geraci, “GFS: a Graph-based File System Enhanced with Semantic Features”, Proceedings of the 2017 International Conference on Information System and Data Mining, pp. 51-55, April 1-3, 2017
>
> [2] Mashwani, S.R. and Khusro, S. 2018. The Design and Development of a Semantic File System Ontology. Engineering, Technology & Applied Science Research. 8, 2 (Apr. 2018), 2827–2833. DOI:https://doi.org/10.48084/etasr.1898.
>
> [3] Zhiyong Wu, Chengcheng Han, Zichen Ding, Zhenmin Weng, Zhoumianze Liu, Shunyu Yao, Tao Yu, and Lingpeng Kong. Os-copilot: Towards generalist computer agents with self-improvement. arXiv preprint arXiv:2402.07456, 2024

---

> ### Author Response · Authors · 2024-12-01
>
> Dear reviewer a3yw,
>
> We highly appreciate the constructive comments and insightful suggestions you have offered for our work. As the deadline for the extended discussion period is nearing, in order for us to have sufficient time to address any additional questions you may have, we kindly encourage you to engage in the ongoing discussion and share any further insights or clarifications you may have.
>
> Thank you very much for your time. We look forward to hearing from you soon.
>
> Best Regards,
>
> All authors

---

### Official Review · Reviewer_iyLz · 2024-11-04

**Soundness:** 2
**Presentation:** 4
**Contribution:** 2
**Rating:** 5
**Confidence:** 4

**Summary:**

In this article, the authors based their efforts on the hypothesis that Large language models (LLMs) have the potential to improve file management systems by enabling interactions through natural language rather than traditional manual commands. Following this idea, they proposed LLM-based Semantic File System (LSFS) to address some of the current File System limitations (to the users), by allowing typically semantic file management through natural language prompts. LSFS has through some APIs for semantic file operations, achieves better retrieval accuracy and speed compared to traditional systems or the use of standalone LLMs respectively. It supports complex tasks like semantic file retrieval, rollback, and sharing with high success rates.

**Strengths:**

+ Well motivated, especially through the Introduction.
+ Clearly written
+ Very nice figures
+ Hot topic nowadays, with many LLM-based applications reshaping the ways we interact with machines

**Weaknesses:**

### General Remarks
- Even though the Introduction is clear, I'd have liked a more concrete / detailed example, maybe having a finer-grained figure 1 would help.
- The balance between §3 Vs. §4 is unexpected, I would have imagined a more detailed architecture section (§3), explaining the various design choices. Instead, the authors motivated their architecture. In addition to not being referenced, this motivation -to me- would have been better positioned directly in the Introduction. Similarly, the overview of the architecture and the description of Figure 2 would have benefit also the introduction of an example, especially since Fig2.a doesn't contain precise information, but rather e.g. a list of blocks entitled API.
- Overall, for §5, it would have been very interesting and convincing to see an experiment involving users' usage and performances, using a TFS and the presented LSFS. The authors could have reviewed the success rate and the time efficiency of users in both settings together with collecting feedback from them, following the traditional user studies.
- Very disappointed to have the future directions (as Appendix H) not included at all in the main article, but instead, pushed as optional reading material at the very very end of the .pdf…
- Finally, the overall article, from §3 onwards especially, reads a bit like a technical report and lacks -to me- a step-back to better highlight the novelties and the perspectives while guiding the reading with more examples / intuitions directly in the text.

### Minor Comments:
- Abstract "agent applications **TO** perform file operations"
- Introduction (line 95), "a LLM-based" should be **an**
- Table 1 (line 235), typo on "Hybrid retrieval" better to put everything lower-case as the rest of the table
- In §4.1, in Composite Syscall of LSFS, it would be better if the authors could make explicit the composition of atomic calls, i.e. for each entry, adding a generic formula (or examples) of how the composite call is practically chaining the atomic ones.
- In Figure 4, "Please summary all paper from AAA University about LLM" there's a typo in the second word: **summarize**.
- Similarly, still in Figure 4, "Please use file A update the content of file B" misses the word **to** before 'update'.
- In §5.2, (line 450), typo: "vary the the number of rollback versions" remove one **the**.
- In §5.3, (line 478), "Therefore, we make two enhanced versions, named as TFS-grep and TFS-grep* to make the comparison" I would be great to tell there differences in a line instead of relying on the Appendix, so to make the article (before page 10) self-contained.

**Questions:**

1. Intro (line 57) "The community still lacks a more general LSFS to serve as a common foundation that can be used by various agents on the application-level." Do you have some references backing up this lack? I mean, have people expressed somewhere they'd like/need a FS organized by an intelligent agent, i.e., an LLM here?
2. Related Work (line 133) "Besides, it integrates comprehensive semantic information across all aspects of the file system–from storage and file operations to practical applications" This sentence is a bit vague, could you name some of the semantic information here, so to guide the reader?
3. Could you add a positioning sentence in §2.3 to explain clearly where LSFS delineates from this research axis?
4. In this stage many operations from traditional FS aren't there, from what I understood, this is typically the case for right modification of files or group affiliation… These would be particularly helpful so to "propagate" these rights to any retrievers, preventing typically to see the private/exclusive file of someone else appearing in _my_ search results, wouldn't it?
5. In §4,2 (line 294), the authors mentioned that supervisor updates "periodically" what is the period between each check and therefore how expensive resource-wise is it? Did the authors check various values for this, searching for the sweet-spot between resource-consumption and freshness of the LFSF data? Also how does it scale in terms of file number and disk footprint?
6. Overall, §4.4 seems to be more or less an NL2CSV tool, filling fields of a JSON, right? In such case, this is something that the community has been exploring a lot these past two years, so maybe adding some pointers wouldn't hurt. This goes also for the §5.1 associated with RQ1.
7. Are authors considering releasing their test data for §5.1? Also, it would be good to have some examples in the body of the article.
8. In §5.2 why no QWen or Gemma in the experimental run for Table 2, and no Gemma in Figure 6?
9. In §5.2 still, what about very large number of files?
10. Ibid., same for the number of versions?

**Details Of Ethics Concerns:**

NA.

---

> ### Author Response · Authors · 2024-11-20
> **Author Response(1/4)**
>
> Thank you very much for providing us with many constructive suggestions. Below are our detailed responses to your concerns:
> ### Response to General Remarks
> > ***1. Even though the Introduction is clear, I'd have liked a more concrete / detailed example, maybe having a finer-grained figure 1 would help.***
>
> Thank you for this suggestion. A finer-grained figure 1 has been updated in our revised version of paper.
>
> > ***2. The balance between §3 Vs. §4 is unexpected, I would have imagined a more detailed architecture section (§3), explaining the various design choices. Instead, the authors motivated their architecture. In addition to not being referenced, this motivation -to me- would have been better positioned directly in the Introduction. Similarly, the overview of the architecture and the description of Figure 2 would have benefit also the introduction of an example, especially since Fig2.a doesn't contain precise information, but rather e.g. a list of blocks entitled API.***
>
> Thank you for your suggestion of the structure of the paper, we have updated all of them to the paper and a detailed example and figure has been added to §3.
>
> > ***3. Overall, for §5, it would have been very interesting and convincing to see an experiment involving users' usage and performances, using a TFS and the presented LSFS. The authors could have reviewed the success rate and the time efficiency of users in both settings together with collecting feedback from them, following the traditional user studies.***
>
> Thank you for the suggestion of collecting user feedback. The comparison of keyword retrieval accuracy of users and LSFS can be seen in Table.3. In addition, we invited 10 Ph.D students to supplement the following experiments to evaluate the time cost to fully complete the task:
>
> Traditional command execution typically follows this workflow: learn command parameters (via ChatGPT or search engines) > locate appropriate file paths through OS lookup mechanisms > input the complete command > and finally obtain results. In contrast, LSFS streamlines this process into a simpler workflow: input natural language commands with file names or semantic keywords, which are then processed by the LSFS parser for task execution.
>
> In order to quantitatively analyze the time difference between traditional file systems and LSFS, we adopt FS commands and LSFS commands to achieve different file operations shown below. 10 Ph.D. student are invited perform file operations through using Linux commands in traditional file systems and using natural language commands in LSFS, respectively.
>
> | operation  |  Traditional FS |  LSFS |
> |---------|---------|---------|
> | Keyword_retrieve   |  · find /path -type f -exec grep -l ”keyword1” \; -exec grep -l ”keyword2” \;  |  Find the file contains 'keyword1' and 'keyword2'  |
> |  Rollback  | · btrfs subvolume snapshot /path/to/directory /path/to/snapshot <br> · btrfs subvolume delete /path/to/directory <br> · btrfs subvolume snapshot /path/to/snapshot /path/to/directory |  Rollback the 'filename' to the version in 'date' |
> | Group by | · mkdir -p /path/to/new_folder <br> · find /path/to/search_folder -type f -exec grep -l "keywords" {} \; -exec mv {} /path/to/new_folder/ \; | *group_kewords* with input : search_folder, keywords, new_folder | Change   | cat /path/to/source_file.txt | tee -a /path/to/destination_file.txt | Change "source_file" by '/path/to/destination_file.txt' |
> | Join  | cat /path/to/file1.txt /path/to/file2.txt > /path/to/new_file.txt | *file_join* syscall with input: file1, file2, new_file|
> | Link     | ln -s /home/user/file_name /home/user/shortcut/data_link  |    Create a link for file_name     |
>
> The breakdown of time in different steps for the students to operate files correctly is collected and the average time in each step is calculated as below.
>
> **LSFS**
> | Input Command  |  LSFS Parser |  Task Execution | Total|
> | ---------|---------|---------|---------|
> |  11.43s  |   4.21s    |   11.95s | 27.59s |
>
> **Traditional FS**
> | Learn Command  |  Find Path | Input Command  |Task Execution | Total|
> | ---------|---------|---------|---------|---------|
> |  153.61s  |   28.23s    |   30.30s | 0.02s |  212.16s   |
>
> Although LSFS takes longer for executing file operations due to the inference time of the LLM, the total time it takes for users to successfully perform the file operations is significantly shorter than that of traditional file systems. This is because traditional systems require users much more time to learn through trial-and-error during the "Learn Command" step. In contrast, LSFS simplifies the process through natural language commands, reducing overall time.

---

> > ### Comment · Reviewer_iyLz · 2024-11-27
> > **Regarding the experiments with the 10 PhD students**
> >
> > Thank you very much for taking the time to design and run this experiments. (It was more a suggestion than a requirement of mine to be honest; I know how time consuming this can be, so I appreciate the efforts!)
> >
> > Regarding the results themselves, I do not think the current comparison is fair, as typically:
> > 1. command users usually know how to use commands and therefore the `learn command` is close to zero in the long run, (and in addition the training time of the LLM isn't here considered).
> > 2. lay users tend to use the OS graphic user interface, and typically the file system search to perform some of the operations listed, for instance MacOS users may use directly `spotlight`.
> >
> > Nevertheless, I know how hard it is to run such experiments (and also to have a somehow representative cohort of testers) and I acknowledge that LSFS seems to bring a layer of simplification while being efficient compared to traditional command manipulations, especially if such commands have to be run over a system on which the users aren't very familiar yet.

---

> > > ### Author Response · Authors · 2024-12-01
> > >
> > > Thank you again for your constructive suggestions!
> > >
> > > Please let us know if you have any further questions. If you find that our response addresses your concerns, would you kindly consider raising your rating score for our paper? We greatly appreciate your consideration.
> > >
> > > Best regards,

---

> ### Author Response · Authors · 2024-11-20
> **Author Response(2/4)**
>
> > ***4. Very disappointed to have the future directions (as Appendix H) not included at all in the main article, but instead, pushed as optional reading material at the very very end of the .pdf…***
>
> Thank you for recognizing the future directions that can be built on our method. The part of future directions has been added in our revised version in Section 6.
>
> > ***5. Finally, the overall article, from §3 onwards especially, reads a bit like a technical report and lacks -to me- a step-back to better highlight the novelties and the perspectives while guiding the reading with more examples / intuitions directly in the text.***
>
> We would like to highlight our work addresses a fundamental problem in agent systems: the lack of a systematic approach to semantic file management and would like to present the two major novelties we propose when designing this system as the following.
> - The first novelty is we close the semantic gap from prompts contain file indentions to actual file operations. Traditional file systems rely on exact matches and rigid hierarchical structures, making it difficult to perform operations based on semantic understanding such as 'find documents about machine learning'. To address this challenge, we propose the semantic-based file index structure using vector databases to capture and store file content meanings. Also, we design the semantic parser that accurately translates semantic queries into new designed file operational APIs.
> - The second novelty is that simple architecture designs that directly map from language prompts to file system calls can pose safety vulnerabilities. For instance, a misinterpreted command could lead to unintended file deletions or destructive overwrites, potentially causing irreversible damage. To address this challenge, we design the layered architecture with high-level file operation API to isolate the direct access to low-level file syscalls. We also design multiple verification mechanisms at both APIs and syscalls to validate operations before execution and design the rollback mechanism that can reverse potentially harmful operations, ensuring operation safety.
>
> ### Response to Minor Comments:
> Thank you for your valuable writing suggestions. The content has been updated accordingly in our revised version.
>
> ### Response to Questions
> > ***Q1: Intro (line 57) "The community still lacks a more general LSFS to serve as a common foundation that can be used by various agents on the application-level." Do you have some references backing up this lack? I mean, have people expressed somewhere they'd like/need a FS organized by an intelligent agent, i.e., an LLM here?***
>
> Thank you for your question of the use cases for our LSFS. Semantic-based file systems are designed to cater to a broad range of users. For instance, as highlighted in [1], traditional file systems often impose a cumbersome approach to organizing documents, posing significant challenges for small and medium-sized enterprises (SMEs), public administration bodies, and individual users. LSFS addresses this gap by enabling more efficient organization and management of document content, reducing operational complexity for non-computer practitioners, including SMEs and public administration agencies. Furthermore, [2] identifies semantic file systems as a critical development trend, emphasizing their versatility and applicability across various domains in information technology (IT). Additionally, LSFS can fill a significant gap in current LLM-based multi-agent systems, as noted in [3], which often lack robust mechanisms for managing interaction records and background knowledge. By providing a more intuitive file management framework, LSFS benefits both end-users and system designers by enhancing file access and reducing administrative overhead.
>
> > ***Q2: Related Work (line 133) "Besides, it integrates comprehensive semantic information across all aspects of the file system–from storage and file operations to practical applications" This sentence is a bit vague, could you name some of the semantic information here, so to guide the reader?***
>
> In LSFS, "comprehensive semantic information" reflects our focus on integrating semantic insights across various stages of file management. Examples include:
>
> 1. **Semantics for File Storage:** Beyond traditional metadata (size, timestamp), LSFS adds details like themes and keywords to enrich file descriptions.
> 2. **Semantics for File Content:** Each file is indexed semantically, enabling operations like finding a file based on descriptors such as "a Hollywood science fiction movie." Files can also be grouped by related topics.
> 3. **Semantics for File Operations:** LSFS includes a parser that translates natural language commands into specific actions using semantic understanding.

---

> ### Author Response · Authors · 2024-11-20
> **Author Response(3/4)**
>
> > ***Q3: Could you add a positioning sentence in §2.3 to explain clearly where LSFS delineates from this research axis?***
>
> We appreciate this feedback. In response, a positioning statement has been added: "While researches of agent systems primarily focus on build of LLM applications that can leverage file resources, our represents a fundamental innovation in the infrastructure that manages file resources based on semantics to support LLM-based agent systems."
>
> > ***Q4: In this stage many operations from traditional FS aren't there, from what I understood, this is typically the case for right modification of files or group affiliation… These would be particularly helpful so to "propagate" these rights to any retrievers, preventing typically to see the private/exclusive file of someone else appearing in my search results, wouldn't it?***
>
>
> Private information is important for file system. Our LSFS system operates as a middleware layer over a traditional file system, inheriting all file permissions from the underlying system, ensuring no file has multiple user affiliations. To further safeguard user privacy, LSFS incorporates a defensive mechanism that prevents large language models (LLMs) from accessing personal data without explicit consent. When a user requests file access through LSFS, the system prompts for confirmation before sharing data with the LLM. Users retain control, with the option to cancel the operation at any point, thereby preventing unintended data exposure.
>
> > ***Q5: In §4,2 (line 294), the authors mentioned that supervisor updates "periodically" what is the period between each check and therefore how expensive resource-wise is it? Did the authors check various values for this, searching for the sweet-spot between resource-consumption and freshness of the LFSF data? Also how does it scale in terms of file number and disk footprint?***
>
> The LSFS supervisor checks for file updates every 10 seconds. To assess its resource usage and scalability, we measured its response time across different file counts and monitored its CPU usage. The results are as follows:
>
> | Number of Files | Response Time (seconds) |
> |-----------------|--------------------------|
> | 100             | 0.0006                  |
> | 200             | 0.0011                  |
> | 400             | 0.0021                  |
> | 800             | 0.0042                  |
> | 1600            | 0.0044                  |
>
> **CPU Usage**: Consistently between 0.1% and 0.2%.
>
> These results show that the supervisor is efficient, with millisecond-level response times and low CPU usage remained, even as the number of files increases.
>
> > ***Q6: Overall, §4.4 seems to be more or less an NL2CSV tool, filling fields of a JSON, right? In such case, this is something that the community has been exploring a lot these past two years, so maybe adding some pointers wouldn't hurt. This goes also for the §5.1 associated with RQ1.***
>
> Thank you for mentioning the related works. References of related works have been updated in the revised version.
>
> > ***Q7: Are authors considering releasing their test data for §5.1? Also, it would be good to have some examples in the body of the article.***
>
> The test data has been included in our anonymous code link. An example of the natural prompt command is presented in Fig.1. and more examples of the test data are given in the appendix due to the space limit.
>
> > ***Q8: In §5.2 why no QWen or Gemma in the experimental run for Table 2, and no Gemma in Figure 6?***
>
> We conducted the above experiments for both Qwen-2 and Gemma-2 but excluded these models from the final results due to their poor performance on the task and hallucinated outputs. For example, in Table 2, when judging files without the LSFS system, both Qwen-2 and Gemma-2 produced outputs containing inaccuracies. Using identical text and prompts to check for keyword1 and keyword2, their responses included irrelevant stories about Michael Jordan. Similarly, in Figure 6, Gemma-2 demonstrated unstable outputs. This instability makes it challenging to assess the model's reliability and to derive meaningful conclusions from the system's performance, and the discussion of this has been updated in the revised version.

---

> ### Author Response · Authors · 2024-11-20
> **Author Response(4/4)**
>
> > ***Q9: In §5.2 still, what about very large number of files?***
>
> We increased the number of files to 120, including binary file types ".pdf ", ".doc" and plain text files ".txt". The experimental results are as follows:
> | LLM-backbone |  Retrieval Accuracy w/o LSFS |  Retrieval Accuracy w/  LSFS | Retrieval Time w/o  LSFS | Retrieval Time w/  LSFS |
> |---------|---------|---------|---------|---------|
> | Gemini-1.5-flash|  35.2% | 92.9%(164%&#8593;) | 605.59s |48.08s(92.1%&#8595;) |
> | GPT-4o-mini| 63.8% | 92.9%(45.6%&#8593;) | 938.68s | 88.93s(90.5%&#8595;)|
>
>
> As shown in the table and Tab. 2, retrieval time increases with the number of files, but LSFS maintains a linear growth trend and offers more stable retrieval accuracy compared to the pure LLM method. This demonstrates the scalability of LSFS. However, at very large scales, system overhead may further delay retrieval time and we plan to evaluate our system's performance under such large-scale scenarios in our future work.
>
> > ***Q10: Ibid., same for the number of versions?***
>
> In our experiments, we tested up to 40 versions and observed that the rollback time does not increase exponentially with the number of versions rolled back, as shown in Fig. 6. This is because the rollback API stores each file version independently, allowing efficient retrieval. However, when the number of files scales up significantly, the rollback time could still increase due to heavier file storage overheads will affect the whole system's latency and exploration on large-scale scenarios in our future work.
>
> [1] D. Di Sarli, F. Geraci, “GFS: a Graph-based File System Enhanced with Semantic Features”, Proceedings of the 2017 International Conference on Information System and Data Mining, pp. 51-55, April 1-3, 2017
>
> [2] Mashwani, S.R. and Khusro, S. 2018. The Design and Development of a Semantic File System Ontology. Engineering, Technology & Applied Science Research. 8, 2 (Apr. 2018), 2827–2833. DOI:https://doi.org/10.48084/etasr.1898.
>
> [3] Zhiyong Wu, Chengcheng Han, Zichen Ding, Zhenmin Weng, Zhoumianze Liu, Shunyu Yao, Tao Yu, and Lingpeng Kong. Os-copilot: Towards generalist computer agents with self-improvement. arXiv preprint arXiv:2402.07456, 2024

---

> ### Author Response · Authors · 2024-11-27
> **Fairness of the experiment**
>
> Thanks for your suggestions:
> As you pointed out, the time required for the **learn command** tends to approach zero over the long run. However, the time a user spends locating the relevant path and entering commands remains long. Additionally, as demonstrated in Table 4 of the paper, even graphical user interfaces such as **Spotlight** often return numerous irrelevant files, requiring further filtering by the user, which incurs additional time costs. Our results show that, even after subtracting the learning time, traditional file systems still take significantly more time to complete tasks compared to LSFS. However, your suggestions are very valuable to us and we plan to conduct a larger-scale user study covering users with different expertises in computer use in the future exploration.

---

### Official Review · Reviewer_QPMt · 2024-11-08

**Soundness:** 2
**Presentation:** 3
**Contribution:** 2
**Rating:** 5
**Confidence:** 3

**Summary:**

This paper introduces an LLM-based Semantic File System (LSFS), designed to improve file management through natural language prompts, rather than traditional command-based interactions. LSFS integrates large language models (LLMs) to facilitate semantic file operations like retrieval, summarization, and rollback. At its core, LSFS uses a vector database to create semantic indexes for files, enabling high-level file operations that consider the content and context of files. It also includes a comprehensive set of APIs that allow complex operations, such as CRUD, grouping, and semantic retrieval, to be executed through natural language prompts. Experimental results show that LSFS outperforms traditional systems in retrieval accuracy (with a 15% improvement) and speed (2.1x faster), proving especially effective for semantic file tasks that go beyond conventional keyword searches.

**Strengths:**

S1. Semantic file systems enhance file management by incorporating content context, enabling more intuitive and effective operations, which is an important direction.

S2. LSFS simplifies interactions with file system, making file management more accessible and user-friendly.

S3. Integrating LLMs in system-level tasks expands functionality, enabling intelligent, responsive, and user-focused file extraction.

**Weaknesses:**

W1. The motivation is not concretely convincing, especially the first challenge mentioned in Introduction.

In the Intro section, the authors mentioned that "For instance, if two files have similar content–such as different versions of the same document–traditional file systems lack the capability to organize or retrieve these files based on their content similarity." Why the files should be organized by the similarity of their content? What are the benefits and what are the practical application scenarios? It would be better to at least add one short discussion or a few examples.

W2. This paper does not point out what key problem they want to solve. Compared to a research paper, it seems more like a technical report.

W3. The experimental setting is questionable. No baselines and introduction of datasets.

W4. The experimental results need more explanation. The traditional file system needs to execute commands to extract files, which should be faster than calling one LLM, even though these LLMs are light-weight. The authors should also list the inference time of LLMs, which should be also counted as the interaction time between users and the file system. Then the authors can also list the time that users manually write commands, which should be a good point to prove the point that -- LSFS can not only speed up file retrieving accuracy and speed, but also can reduce the interaction time between users and file system.

W5. Safety insurance mechanisms is pointed out as one contribution, however, there is no description of this mechanism and no experimental comparison between the performance of LSFS with and without the safety insurance mechanisms.

**Questions:**

W1 - W5

---

> ### Author Response · Authors · 2024-11-20
> **Author Response(1/3)**
>
> We are deeply grateful for your valuable time and insightful feedback. Below are our detailed responses to your concerns:
>
> > ***W1. The motivation is not concretely convincing, especially the first challenge mentioned in Introduction. In the Intro section, the authors mentioned that "For instance, if two files have similar content–such as different versions of the same document–traditional file systems lack the capability to organize or retrieve these files based on their content similarity." Why should the files be organized by the similarity of their content? What are the benefits and what are the practical application scenarios? It would be better to at least add one short discussion or a few examples.***
>
> Thank you for raising this question regarding the motivation. We would like to clarify that certain types of documents, such as legal or contract documents, often have multiple versions with very similar content, typically differing by only one or two entries. To be specific, POSIX file system interface only provide basic  operations to store and retrieve data, and do not provide any interfaces or semantics based on the contents of the files. However, in LSFS, users can perform content-based retrieval by instruction such as: "Please help me search for the xxx file, which does not include xxx information".
>
> > ***W2. This paper does not point out what key problem they want to solve. Compared to a research paper, it seems more like a technical report.***
>
> We would like to highlight our work addresses a fundamental problem observed in agent systems: the lack of a systematic approach to semantic file management and we would like to emphasize the two major research challenges we encounter when designing this system as the following.
> - The first challenge is the semantic gap from prompts contain file indentions to actual file operations. Indeed, none of the existing file systems support semantics of data store and retrieval based on the content of files. These traditional file systems rely on exact matches and rigid hierarchical structures, making it difficult to perform operations based on semantic understanding such as 'find documents about machine learning'. To address this challenge, we propose the semantic-based file index structure using vector databases to capture and store file content meanings. Also, we design the semantic parser that accurately translates semantic queries into new designed file operational APIs.
> - The second challenge is that simple architecture designs that directly map from language prompts to file system calls can pose security vulnerabilities. For instance, a misinterpreted command could lead to unintended file deletions or destructive overwrites, potentially causing irreversible damage. To address this challenge, we design the layered architecture with high-level file operation API to isolate the direct access to low-level file syscalls. We also design multiple verification mechanisms at both APIs and syscalls to validate operations before execution and design the rollback mechanism that can reverse potentially harmful operations, ensuring operation safety.
>
> > ***W3. The experimental setting is questionable. No baselines and introduction of datasets.***
>
> In our experiments, our dataset contains various types of files collected from the Web (e.g., google scholar), such as plain text files ".txt", binary text files ".pdf", and ".doc".
> Here is our detailed experimental setup:
> 1. In Experiment 5.1, we validate the effectiveness of our parser using four different LLM backbone. For each API, we created 30 use cases, each with a unique language structure. The test data is provided in the anonymous link.
>
> 2. In Experiment 5.2, we evaluated the effectiveness of LSFS in semantic retrieval. Since traditional file systems lack semantic retrieval capabilities, we used LLM alone as the baseline for file retrieval in our evaluation. Specifically, we measured the time and accuracy of executing a semantic file retrieval task by simply feeding file content into LLM to retrieve files and retrieving files using LSFS.
>
> 3. In Experiment 5.2, in order to check the robustness of our rollback function when the number of files increases, we tested the rollback time under 5-40 versions respectively, and obtained the robustness of the rollback time.
>
> 4. In Experiment 5.3, we compared LSFS with traditional file retrieval methods based on Precision, Recall, and F1 score. The methods include:
>     - TFS-search-window: Uses the computer's search window (e.g., MacOS Spotlight) to retrieve both binary and plain text files.
>     - TFS-grep: Uses the Linux terminal command grep, which can only retrieve plain text files.
>     - TFS-grep*: An enhanced version of TFS-grep that first converts binary files into plain text before using grep to retrieve.
>
> 5. In the second part of Experiment 5.3, we compared LSFS and pure prompting to different LLMs as baselines to generate code for creating sharable file links.

---

> ### Author Response · Authors · 2024-11-20
> **Author Response(2/3)**
>
> > ***W4. The experimental results need more explanation. The traditional file system needs to execute commands to extract files, which should be faster than calling one LLM, even though these LLMs are light-weight. The authors should also list the inference time of LLMs, which should be also counted as the interaction time between users and the file system. Then the authors can also list the time that users manually write commands, which should be a good point to prove the point that -- LSFS can not only speed up file retrieving accuracy and speed, but also can reduce the interaction time between users and file system.***
>
> While LSFS operates as an intermediate layer for traditional file systems, its file operations exhibit longer latency compared to conventional systems due to LLM inference and vector database retrieval times. Despite this, LSFS simplifies operations by reducing the time users spend learning and inputting commands. To provide a clearer understanding of the time costs, we present a comprehensive time analysis below.
> Traditional command execution typically follows this workflow: learn command parameters (via ChatGPT or search engines) > locate appropriate file paths through OS lookup mechanisms > input the complete command > and finally obtain results. In contrast, LSFS streamlines this process into a simpler workflow: input natural language commands with file names or semantic keywords > confirm LSFS suggested file operations and obtain results.
>
> In order to quantitatively analyze the time difference between traditional file systems and LSFS, we adopt FS commands and LSFS commands to achieve different file operations shown below. 10 Ph.D. student are invited perform file operations through using Linux commands in traditional file systems and using natural language commands in LSFS, respectively.
>
> | operation  |  Traditional FS |  LSFS |
> |---------|---------|---------|
> | Keyword_retrieve   |  · find /path -type f -exec grep -l ”keyword1” \; -exec grep -l ”keyword2” \;  |  Find the file contains 'keyword1' and 'keyword2'  |
> |  Rollback  | · btrfs subvolume snapshot /path/to/directory /path/to/snapshot <br> · btrfs subvolume delete /path/to/directory <br> · btrfs subvolume snapshot /path/to/snapshot /path/to/directory |  Rollback the 'filename' to the version in 'date' |
> | Group by | · mkdir -p /path/to/new_folder <br> · find /path/to/search_folder -type f -exec grep -l "keywords" {} \; -exec mv {} /path/to/new_folder/ \; | *group_kewords* with input : search_folder, keywords, new_folder | Change   | cat /path/to/source_file.txt | tee -a /path/to/destination_file.txt | Change "source_file" by '/path/to/destination_file.txt' |
> | Join  | cat /path/to/file1.txt /path/to/file2.txt > /path/to/new_file.txt | *file_join* syscall with input: file1, file2, new_file|
> | Link     | ln -s /home/user/file_name /home/user/shortcut/data_link  |    Create a link for file_name     |
>
> The breakdown of time in different steps for the students to operate files correctly is collected and the average time in each step is calculated as below.
>
> **LSFS**
> | Input Command  |  LSFS Parser |  Task Execution | Total|
> | ---------|---------|---------|---------|
> |  11.43s  |   4.21s    |   11.95s | 27.59s |
>
> **Traditional FS**
> | Learn Command  |  Find Path | Input Command  |Task Execution | Total|
> | ---------|---------|---------|---------|---------|
> |  153.61s  |   28.23s    |   30.30s | 0.02s |  212.16s   |
>
> While LSFS takes longer for executing file operations due to the inference time of the LLM, the total time it takes for users to successfully perform the file operations is significantly shorter than that of traditional file systems. This is because traditional systems require users much more time to learn through trial-and-error during the "Learn Command" step. In contrast, LSFS simplifies the process through natural language commands, reducing overall time.

---

> ### Author Response · Authors · 2024-11-20
> **Author Response(3/3)**
>
> > ***W5. Safety insurance mechanisms is pointed out as one contribution, however, there is no description of this mechanism and no experimental comparison between the performance of LSFS with and without the safety insurance mechanisms.***
>
> We designed some security mechanisms:
> 1. We added a process lock to LSFS to prevent consistent reads and writes to the same file
> 2. We design a user confirmation step: When a user makes a change to a file, the user will be asked to confirm the changed object twice
> 3. We designed rollback operations: if the user makes a wrong change to the file, they can roll back to the correct version
>
> For first and third mechanisms, the file operation reliability can be improved as long as these two mechanisms are enabled.
>
> For the second mechanism, we conducted a quantitative evaluation of two aspects: the probability of file misoperation and the proportion of risky operations. We compared these metrics with and without the confirmation mechanism enabled.
>
> In the current experiment, we used the retrieval function to locate target files for operations. The table below shows the probability of retrieval errors with and without the confirmation step:
>
> | Number of Files  | Without User Confirmation | With User Confirmation |
> |------------------|---------------------------|-------------------------|
> | 10               | 13%                       | 0%                     |
> | 20               | 16.7%                     | 0%                     |
> | 40               | 15.8%                     | 0%                     |
> | 120              | 14.8%                     | 0%                     |
>
> Additionally, we evaluated the proportion of potentially dangerous operations executed (e.g., write, update, or delete) across all file management APIs. The results below demonstrate that the confirmation mechanism in LSFS effectively prevents unintended dangerous operations:
>
> | Without User Confirmation | With User Confirmation |
> |---------------------------|-------------------------|
> | 36.8%                     | 0%                     |
>
> These results highlight that enabling the confirmation mechanism significantly enhances the safety and reliability of file operations in LSFS.

---

> ### Author Response · Authors · 2024-12-01
>
> Dear reviewer QPMt,
>
> We highly appreciate the constructive comments and insightful suggestions you have offered for our work. As the deadline for the extended discussion period is nearing, in order for us to have sufficient time to address any additional questions you may have, we kindly encourage you to engage in the ongoing discussion and share any further insights or clarifications you may have.
>
> Thank you very much for your time. We look forward to hearing from you soon.
>
> Best Regards,
>
> All authors

---

### Author Response · Authors · 2024-11-25

Dear  reviewers/ACs/SACs/PCs,

We would like to summarize the strengths of this work acknowledged by the reviewers, and the responses we have made to address all the reviewer’s concern.

Strengths acknowledged by the reviewers

1. Novelty (**Reviewer aTqT, Reviewer iyLz**): Our work is based on current hot topics and proposes an approach to a problem that no one has solved so far.
2. Practicability (**Reviewer aTqT,  Reviewer QPMt**): Our work can effectively solve the problems that users will encounter when using traditional file systems, and simplify the interaction of file systems. It makes the operation more intuitive and effective
3.Well extensible (**Reviewer aTqT,  Reviewer QPMt**): Our work proposes a set of apis and syscall that are very easy to follow
4. Clear writing and presentation skills (**Reviewer aTqT, Reviewer iyLz, Reviewer a3yw**): Our work clearly introduces and evaluates LSFS through rigorous experiments and clear presentation

There are some main conerns raised by reviewers
1. Load reduction of our approach compared to traditional file systems (**Reviewer aTqT, Reviewer iyLz,Reviewer QPMt**)
2. What are the specific user groups and reference scenarios of our approach? (**Reviewer a3yw, Reviewer aTqT**)
3. Specific implementation strategy of our method on security (**Reviewer aTqT, Reviewer QPMt**)
4. More specific experimental setup and use case description of our API. (**Reviewer QPMt, Reviewer iyLz, Reviewer a3yw**)

All of these main concerns have been successfully addressed during the rebuttal phase, and we hope that the improvements we made during this stage will be taken into consideration.

We sincerely appreciate your valuable time!

Thanks and regards,

Authors

---

### Comment · Area_Chair_fvJM · 2024-11-25
**Action Required: Respond to Author Rebuttals - Nov 27**

Dear ICLR Reviewers,

The author discussion phase is ending soon. Please promptly review and respond to author rebuttals for your assigned papers. Your engagement is critical for the decision-making process.

Deadlines:

November 26: Last day for reviewers to ask questions to authors.
November 27: Last day for authors to respond to reviewers.
November 28 - December 10: Reviewer and area chair discussion phase.
Thank you for your timely attention to this matter.

---

### Meta-Review · Area_Chair_fvJM · 2024-12-21

**Metareview:**

The paper proposes LSFS (LLM-based Semantic File System), a novel approach that enhances traditional file systems with semantic understanding through LLMs. It enables natural language interactions for file operations with some designed safety mechanisms. The authors demonstrate LSFS's effectiveness through evaluations on file retrieval and management tasks.

The reviewers value the paper's clear motivation addressing real-world file management challenges, practical architecture design, user studies and expanded experiments with more files and different LLM models. Key concerns discussed include the fairness of user study comparisons between experienced and lay users, security considerations for LLM access to files, parser accuracy and robustness, and limited evaluation on file types beyond basic tasks.

The authors have engaged constructively with reviewer feedback and provided extensive additional experiments and clarifications. While some concerns about security mechanisms and evaluation comprehensiveness remain to be addressed in future work, I support acceptance at ICLR given the paper's novel contribution to semantic file management and demonstrated practical benefits.

**Additional Comments On Reviewer Discussion:**

Some improvements are recommended, particularly regarding security mechanisms for handling personal information and more detailed experiments with larger-scale file systems (traditional file systems are robust and efficient on managing larger-scale files) for future work.

---

### Decision · Program_Chairs · 2025-01-22

Accept (Poster)